# AANet: Virtual Screening under Structural Uncertainty via Alignment and Aggregation

**Wenyu Zhu**[1*], **Jianhui Wang**[1,5*], **Bowen Gao**[1,4*], **Yinjun Jia**[1], **Haichuan Tan**[1,4],
**Ya-Qin Zhang**[1], **Wei-Ying Ma**[1], **Yanyan Lan**[1,2,3†]

[1]Institute for AI Industry Research, Tsinghua University, Beijing, China
[2]Beijing Frontier Research Center for Biological Structure, Tsinghua University, Beijing, China
[3]Beijing Academy of Artificial Intelligence (BAAI), Beijing, China
[4]Department of Computer Science and Technology, Tsinghua University, Beijing, China
[5]University of Electronic Science and Technology of China, Chengdu, China

## Abstract

Virtual screening (VS) is a critical component of modern drug discovery, yet most existing methods—whether physics-based or deep learning-based—are developed around *holo* protein structures with known ligand-bound pockets. Consequently, their performance degrades significantly on *apo* or predicted structures such as those from AlphaFold2, which are more representative of real-world early-stage drug discovery, where pocket information is often missing. In this paper, we introduce an alignment-and-aggregation framework to enable accurate virtual screening under structural uncertainty. Our method comprises two core components: (1) a tri-modal contrastive learning module that aligns representations of the ligand, the *holo* pocket, and cavities detected from structures, thereby enhancing robustness to pocket localization error; and (2) a cross-attention based adapter for dynamically aggregating candidate binding sites, enabling the model to learn from activity data even without precise pocket annotations. We evaluated our method on a newly curated benchmark of *apo* structures, where it significantly outperforms state-of-the-art methods in blind apo setting, improving the early enrichment factor (EF1%) from 11.75 to 37.19. Notably, it also maintains strong performance on *holo* structures. These results demonstrate the promise of our approach in advancing first-in-class drug discovery, particularly in scenarios lacking experimentally resolved protein-ligand complexes. Our implementation is publicly available at `https://github.com/Wiley-Z/AANet`.

## 1 Introduction

Virtual screening (VS) is a cornerstone of modern drug discovery, enabling fast and cost-effective identification of potential small-molecule binders from large chemical libraries. Among various strategies, structure-based virtual screening (SBVS) is particularly prominent, using either physics-based docking [1, 2] or deep learning methods [3] to evaluate compound–pocket compatibility, typically on experimentally resolved *holo* structures. However, most well-characterized *holo* targets have already been explored, limiting discovery opportunities. Recent advances in protein structure prediction, notably AlphaFold2 [4], have dramatically expanded structural coverage, enabling SBVS to target previously inaccessible proteins and supporting early-stage discovery of first-in-class therapeutics.

---

[*]Equal contirbution
[†]Correspondence to `lanyanyan@air.tsinghua.edu.cn`

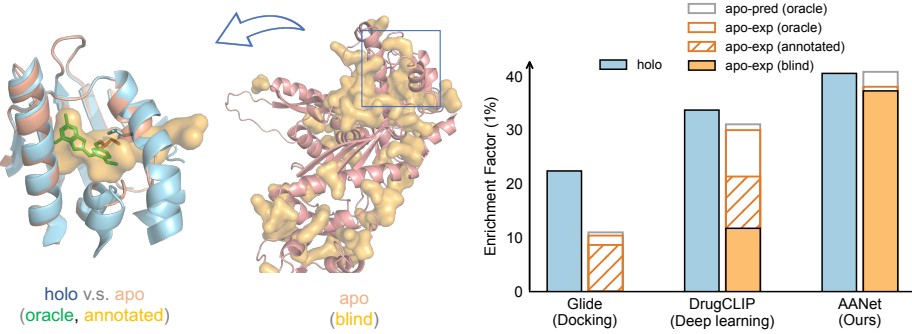

Figure 1: Performance comparison under *holo* and *apo* settings. The bar for docking in the *apo* (blind) setting is absent due to the high computational cost.

However, adapting existing SBVS methods to predicted structures remains a significant challenge. Prior studies [5, 6, 7, 8] have shown that docking performance degrades substantially in predict structures. To address this, flexible refinement [9] and flexible docking techniques [10, 11, 12, 13] have been proposed to adjust local pocket geometries and better accommodate ligands. However, these methods typically rely on pocket annotations derived from *holo* structure-an assumption that does not hold in realistic *apo* or predicted settings. Consequently, these methods largely overlook the upstream challenge of binding site identification under structural uncertainty—a critical bottleneck that severely limits docking performance.

To systematically investigate this problem, we curated a benchmark derived from DUD-E [14] and LIT-PCBA [15], where *holo* protein structures are replaced by *apo* structures, including both predicted and experimentally determined conformations. Candidate pockets were identified using the widely adopted detection tool Fpocket [16]. Our comprehensive evaluation on this benchmark reveals that while deep learning methods such as DrugCLIP [3] exhibit greater robustness to local conformational variation than traditional docking approaches, they still suffer significant performance drops when applied to fully predicted *apo* structures. These findings indicate that effectively modeling and learning the discrepancies between pockets predicted by detection software and the actual ligand-binding sites is a crucial challenge for successful virtual screening in *apo* settings.

To address this challenge, we propose **AANet**, an **Alignment**-and-**Aggregation** framework to improve virtual screening under structural uncertainty in **Apo** and **AlphaFold** (predicted) structures. The first component, alignment, is implemented as a tri-modal contrastive learning scheme, leveraging the insight that pocket detection tools identify geometric cavities, while *holo*-structure represent actual ligand-binding regions. Our method takes three inputs—the ligand, the *holo* pocket, and the detected cavity—and learns pairwise alignments through contrastive objectives. This alignment encourages the model to learn robust and transferable representations across structural discrepancies. To enhance this process, we incorporate a hard negative sampling strategy among candidate cavities, forcing the model to distinguish true binding sites from geometrically plausible but functionally-irrelevant pockets. Building upon this pocket-aware alignment, the second component, aggregation, employs a cross-attention adapter module for dynamic integration of information across multiple candidate cavities. This allows the model to softly weigh pocket representations, infer binding-relevant regions and effectively leverage pocket-agnostic activity data.

Our framework outperforms both physics-based and DL-based baselines, achieving near-holo performance on both predicted and experimental *apo* structures. This improvement is supported by strong pocket identification accuracy and robustness across different pocket detection algorithms, suggesting that the model captures spatial features intrinsic to the structure rather than overfitting to a specific detector. These results highlight the potential of our approach to enable SBVS in more realistic and structurally uncertain drug discovery scenarios, especially where *holo* structures are unavailable.

In summary, our contributions are as follows:

(1) **Revealing the bottlenecks of SBVS under structural uncertainty.** We formalize the problem of virtual screening without reliable pocket definitions, and introduce a benchmark based on DUD-E

and LIT-PCBA for systematic evaluation on predicted and experimental *apo* structures. Our analysis shows that degradation mainly arises from pocket mislocalization, rather than structural noise.

(2) **Alignment and aggregation for uncertain-pocket SBVS.** We propose a tri-modal contrastive learning framework that aligns ligands with geometry-derived cavities via cavity-based augmentation and hard negative mining. A cross-attention adapter further aggregates signals across candidate pockets, enabling training on pocket-agnostic data.

(3) **Enabling virtual screening beyond *holo* structures.** Our method achieves performance under structural uncertainty comparable to that on *holo* structures, enabling screening on targets without annotated binding sites and expanding the reach of structure-based drug discovery.

## 2 Related work

Traditional docking methods such as AutoDock [2] and Glide [1] rely on physics-based scoring functions to evaluate target-ligand interactions. A range DL-based approaches have emerged that learn scoring functions from protein–ligand poses [17, 18, 19, 20, 21, 22] or predict interactions based on structural inputs [23, 24]. Recent methods such as DrugCLIP [3] adopt a novel contrastive paradigm inspired by CLIP, aligning ligands and protein pockets in a shared embedding space, thus representing a new direction in SBVS.

Nonetheless, most of these methods assume access to high-quality *holo* structures and overlook the practical challenges posed by predict *apo* proteins. The accuracy of conventional docking deteriorates significantly in these settings due to the lack of ligand-induced conformational changes [6, 7, 8], although some approaches attempt to mitigate this through flexible modeling [9] or by leveraging homologous *holo* structures [5]. However, DrugCLIP and other DL-based methods exhibits robustness to structural perturbations; Instead, our analysis suggests that it is highly sensitive to the location and quality of the predefined binding pocket in *holo* structures.

## 3 Method

### 3.1 Formulating virtual screening under structural uncertainty

SBVS aims to identify bioactive molecules from a candidate library $\mathcal{M} = \{m_1, m_2, \ldots, m_n\}$, given a protein structure $x_n \in \mathbb{R}^{3 \times N}$. In conventional settings, the protein is provided in *holo* form, and the binding pocket $P_l$ is defined by the spatial neighborhood surrounding a co-crystallized ligand $x_m \in \mathbb{R}^{3 \times M}$:

$$P_l = \left\{ x_n \in \mathbb{R}^3 \ \middle| \ \min_{m \in \{1, \ldots, M\}} \|x_n - x_m\| \leq d \right\} \tag{1}$$

where $d$ is typically set to 6 Å. This *holo* pocket allows physics-based docking methods or learning-based scoring functions to estimate binding affinity.

However, *holo* structures are often unavailable in realistic drug discovery pipelines. Instead, predicted or experimental *apo* structures are used, in which the protein has not undergone ligand-induced conformational changes. In such cases, where ligand positions are unknown, binding pockets must be inferred solely from the protein structure. To address this, various pocket detection tools have been developed [16, 25, 26, 27, 28]. They identify potential binding sites by characterizing geometric cavities on the protein surface. Specifically, let $\{x_c^{(s)}\}_{s=1}^{S}$ be the centers of $S$ cavities detected by pocket prediction software. A candidate pocket is then defined as:

$$P_c^{(s)} = \left\{ x_n \in \mathbb{R}^3 \ \middle| \ \|x_n - x_c^{(s)}\| \leq d \right\} \tag{2}$$

Given an *apo* protein structure $x_n$ and a set of candidate pockets $P_c^{(s)}$ identified via geometric cavity detection, the problem of structure-based virtual screening (SBVS) under structural uncertainty is to accurately identify bioactive molecules from the compound library $\mathcal{M}$.

### 3.2 Disentangling failure modes in virtual screening without holo structures

While prior studies [6, 7, 8] have reported significant performance drops when using *apo* or predicted structures, their focus has largely been on traditional docking methods. In this work, we systematically

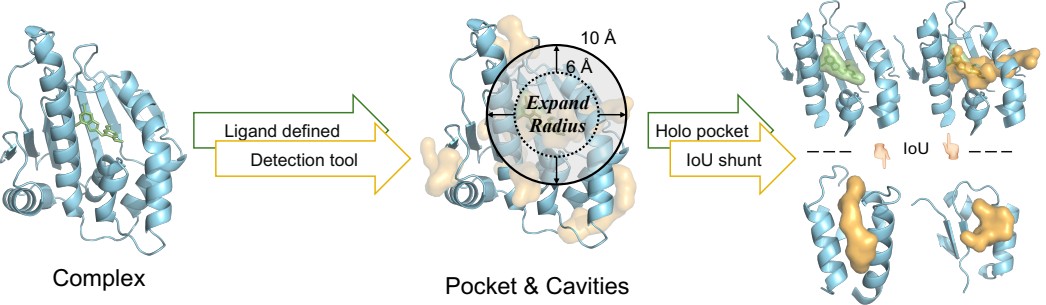

Figure 2: Cavity-based pocket augmentation with hard negative mining. For each protein–ligand complex, the *holo* pocket is defined by the ligand, and a pocket detection tool scans cavities on the protein structure. Cavities are labeled positive or negative based on their IoU with the *holo* pocket.

evaluate both docking-based and deep learning (DL) approaches, identifying the key challenges that limit performance under structural uncertainty.

We curated a benchmark from the DUD-E dataset [14] to assess SBVS under both structural and pocket-level uncertainty. For each target, we collected a matched set of experimentally determined *holo* and *apo* structures, as well as AlphaFold2-predicted models [9], enabling controlled comparisons across structure types under consistent ligand–target pairs. A total of 38 targets with all three structure types were retained. Compared to *holo* structures, *apo* and predicted proteins lack ligand-induced conformational changes, introducing two types of uncertainty: (i) *structure mismatch*, where the backbone conformation differs but pocket location is assumed fixed; and (ii) *pocket localization mismatch*, where the true binding site must be inferred due to missing ligand supervision. To isolate these effects, we define evaluation settings along two axes. From the structure perspective, we consider **apo-exp** (experimental *apo*) and **apo-pred** (AlphaFold2). For pocket localization, we evaluate: **oracle**, where the ligand-defined pocket is used; **annotated**, where the detected cavity with highest IoU to the true pocket is selected; and **blind**, where no ligand or annotation is available and pockets are detected purely from geometry.

We evaluated Glide and the DL-based DrugCLIP [3] across these settings. As shown in Figure 1 and Table 1, Glide shows a sharp drop even in the apo-exp (oracle) setting, revealing its sensitivity to conformational changes. In contrast, DrugCLIP maintains performance across *holo*, apo-exp, and apo-pred (oracle), indicating robustness to moderate structural noise. However, its performance declines significantly in the annotated and blind settings, where pocket localization is uncertain. Even slight misalignments in pocket position lead to notable degradation (see Appendix A.1).

These findings expose a key limitation of current DL-based SBVS methods: while tolerant to structural variation, they remain dependent on accurate pocket definitions. Without them, performance drops sharply—highlighting the need for models that can infer relevant pockets under uncertainty. In the next section, we introduce a contrastive learning framework that addresses this challenge through cavity alignment and multi-pocket aggregation.

### 3.3 Tri-modal contrastive alignment

We propose a tri-modal contrastive alignment framework that aligns representations across ligand, *holo* pocket, and cavity modalities to overcome localization-induced failures. The key idea is to disentangle pocket representations from their dependence on ligand-defined positions and encourage alignment with geometry-derived cavities.

**Cavity-based pocket augmentation via proxy selection.** Due to the performance loss caused by pocket deviation, we introduced a pocket-side augmentation that aligns both *holo* ligand and *holo* pocket with cavity intrinsically residing on the protein structure. To be specified, we adapt Fpocket [16] to detect potential pockets, and randomly select one candidate cavity $P_c$ whose overlap with the *holo* pocket $P_l$ exceeds a predefined threshold $\tau$, measured by Intersection over Union (IoU):

$$IoU(P_l, P_c) = \frac{|P_l \cap P_c|}{|P_l \cup P_c|}. \tag{3}$$

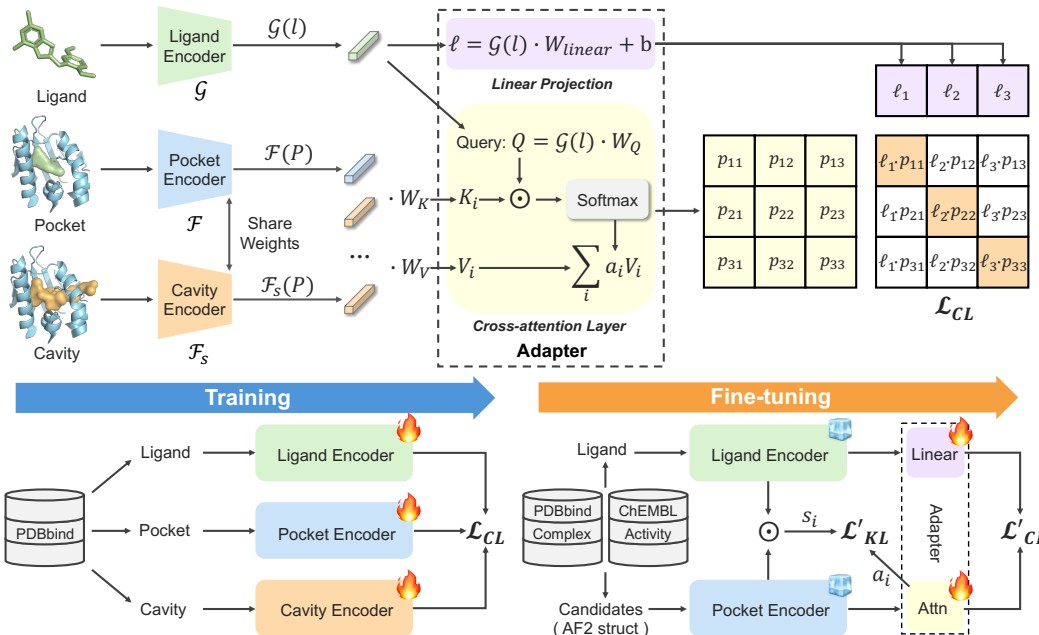

Figure 3: Model framework. AANet operates in two phases: alignment and aggregation. During alignment, representations of the ligand, *holo* pocket, and cavity—encoded separately—are aligned via contrastive losses. In the aggregation phase, the encoders are frozen, and a cross-attention module aggregates representations from candidate cavities (via the cavity encoder) using the ligand embedding as the query. This phase is trained on AlphaFold2-predicted structures without pocket annotations. The ligand embedding is further projected through a trainable linear layer, and a final contrastive loss aligns the adapted ligand and aggregated cavity representations.

**Contrastive objective.** The detected cavity $P_c$ is treated as a third modality alongside the *holo* pocket $P_l$ and the ligand $l$. Let $\mathcal{F}$ and $\mathcal{G}$ denote the encoders for *holo* pockets and ligands, respectively, and let $\mathcal{F}_s$ (sharing weights with $\mathcal{F}$) encode detected cavities. We define two positive pairwise-sigmoid loss [29] functions: one for pocket–ligand pairs $(P, l)$, and one for pocket–pocket pairs $(P_c, P_l)$:

$$\mathcal{L}_{p,l}(P,l) = \log\big(1 + e^{-t\,\mathcal{F}(P)\cdot\mathcal{G}(l)+b}\big), \quad \mathcal{L}_{p,p}(P_c,P_l) = \log\big(1 + e^{-t\,\mathcal{F}_s(P_c)\cdot\mathcal{F}(P_l)+b}\big), \quad (4)$$

where "·" is the dot product, $t > 0$ is a learnable temperature, and $b$ is a learnable bias.

Combining these, the positive alignment objective becomes:

$$\mathcal{L}_{\mathrm{CL}} = \mathcal{L}_{p,l}\big(\mathcal{F}(P_l), \mathcal{G}(l)\big) + \mathcal{L}_{p,l}\big(\mathcal{F}_s(P_c), \mathcal{G}(l)\big) + \mathcal{L}_{p,p}\big(\mathcal{F}_s(P_c), \mathcal{F}(P_l)\big). \quad (5)$$

Aligning both the *holo* pocket and the *holo* ligand with the detected pocket modality mitigates overfitting caused by ligand-dependent pocket extraction. This encourages the model to learn intrinsic spatial features of the protein structure rather than artifacts closely tied to the *holo* ligand's location.

In addition, given the observation that the detected pockets can substantially deviate from *holo* pockets, especially when cramped regions within a pocket cause the detected cavity to split into smaller sub-pockets (see Appendix A.2), we increased the pocket extraction radius from 6 Å to 10 Å while maintaining the same maximum number of atoms by applying atom down-sampling to mitigate this issue.

**Hard negative mining from non-binding cavities.** We then complement the positive terms with negative sampling to equip the model with the ability to distinguish true binding pockets from a set of candidate cavities. We introduce negative samples by selecting non-binding detected pockets during training. Specifically, half of the detected cavities with low IoU are treated as non-binding $P_c^-$ and used as negative pairs under the same pairwise-sigmoid loss (Equation 4) with label $z = -1$ for both $(P_c^-, l)$ and $(P_c^-, P_l)$.

### 3.4 Cross-attention Adapter for Cavity Aggregation

Given that AANet has learned to distinguish binding from non-binding cavities via contrastive supervision, we extend its applicability to activity datasets where the true binding site is unknown. To enable dynamic inference over multiple candidate cavities, we introduce a lightweight **cross-attention adapter** that aggregates cavity embeddings conditioned on the ligand representation.

**Cross-attention design.** The adapter consists of a single-head dot-product attention layer on the cavity side and a linear projection on the ligand side. Given ligand embedding $\mathcal{G}(l)$ and cavity embeddings $\{\mathcal{F}_s(P_c^{(s)})\}_{s=1}^S$, the adapter computes a unified cavity representation:

$$\tilde{e}_c = \sum_{s=1}^{S} a^{(s)} \cdot \mathcal{F}_s(P_c^{(s)}), \tag{6}$$

where attention weights $a^{(s)}$ are computed with the ligand as query and cavity embeddings as keys and values. The ligand is projected to $\tilde{e}_l$ and used in contrastive alignment with $\tilde{e}_c$.

**Initialization.** We initialize the adapter near identity: cavity embeddings are initially averaged, and the ligand embedding remains unchanged. The attention temperature is set high so that softmax approximates a hard max, smoothly transitioning from ensemble scoring to learnable attention during fine-tuning.

**Training with pocket-agnostic data.** To enable training without known binding annotations, we adopt two stabilization strategies: (1) retain a subset of complex-based samples with known binding pockets; and (2) supervise attention weights using soft or hard labels depending on the data source. For activity-only samples, we use the pretrained AANet's cavity scores as soft labels. For complex-based samples, we apply one-hot labels based on ground-truth or high-IoU cavities. Attention supervision is implemented via KL divergence:

$$\mathcal{L}'_{\mathrm{KL}} = \sum_i s_i \log \frac{s_i}{a_i}, \tag{7}$$

where $s$ is either a soft distribution or a one-hot vector.

We also define a contrastive loss between the aggregated cavity embedding $\tilde{e}_c$ and projected ligand embedding $\tilde{e}_l$, using the pairwise-sigmoid form from Equation 4:

$$\mathcal{L}'_{\mathrm{CL}} = \log\left(1 + e^{-t\,\tilde{e}_c \cdot \tilde{e}_l + b}\right), \tag{8}$$

where $t$ and $b$ are shared with the pretraining objective.

The final training objective combines both terms:

$$\mathcal{L}_{\mathrm{agg}} = \mathcal{L}'_{\mathrm{CL}} + \lambda \cdot \mathcal{L}'_{\mathrm{KL}}, \tag{9}$$

allowing the model to infer binding-relevant cavities through dynamic cross-attention, even in the absence of explicit structural annotations.

## 4 Experiments

### 4.1 Experimental settings

**Training.** Our model was initialized with Uni-Mol [30] and fine-tuned on the PDBBind 2020 general set [31], with all entities overlapping with DUD-E or LIT-PCBA removed. For dynamic aggregation training, binding and activity data from ChEMBL35 [32] (pre-processing described in B.2) were filtered and mapped to predicted *apo* structures from the AlphaFold Protein Structure Database [33]. All UniProt entries matching any PDB entity in DUD-E or LIT-PCBA were excluded to avoid data leakage. The model directly predicts binding scores from protein–ligand pairs and is evaluated in a virtual screening setting without any task-specific fine-tuning.

**Evaluation dataset.** We evaluated our model on two virtual screening datasets: DUD-E [14] and LIT-PCBA [15]. The target list was curated to ensure fair comparison across structural sources under controlled conditions. For DUD-E, we used a subset of 38 targets for which all three structure types are available–experimentally resolved *apo* structures, AlphaFold2-predicted structures, and

*holo* structures from the original DUD-E dataset. For LIT-PCBA, experimental *apo* structures were manually selected with the assistance of AHoJ [34], and predicted *apo* structures were obtained from the AlphaFold Protein Structure Database [33]. Twelve targets were retained, and three were excluded: VDR and OPRK1 due to the absence of experimental *apo* structures, and mTORC1 because all *holo* ligands bind at a heterogeneous protein–protein interface, which is not represented in the AlphaFold Protein Structure Database. The evaluation settings follow those described in Section 3.2. The COACH420 dataset for pocket identification was obtained from the P2Rank [28] repository and deduplicated against PDBbind.

**Metrics.** To assess screening performance, we adopted standard virtual screening metrics. AUROC (Area Under the Receiver Operating Characteristic Curve) measures overall ranking performance across all thresholds. EF1% (Enrichment Factor at 1%) measures the fold increase in the proportion of actives within the top 1% of ranked compounds compared to the full dataset, reflecting early recognition ability. BEDROC (with $\alpha = 80.5$) is a weighted variant of the AUROC that emphasizes early enrichment while accounting for overall ranking quality. All metrics were computed per target and averaged across tasks to assess model robustness.

**Baselines.** For docking-based baselines, we included Glide [1] (Standard Precision) and rescoring method RTMScore [22] and EquiScore [21]. Only the top-1 pose per ligand from Glide was retained for fair comparison. We also included TankBind [23] and DrugCLIP [3] as docking-free baselines.

### 4.2 Screening from holo to apo settings

Table 1: Performance on **DUD-E** and **LIT-PCBA**. Each method is evaluated on three structural subsets: **holo**, **apo (experimental)**, and **apo (predicted)**, under both annotated and blind settings. Bold numbers indicate the best performance in each dataset–subset configuration. Row colors indicate method type: Docking & Rescoring , Docking-free baseline , and Proposed method .

| Method | BEDROC ($\alpha = 80.5$) | | | | | EF1% | | | | |
|---|---|---|---|---|---|---|---|---|---|---|
| | holo | apo-exp (annot) | apo-pred (annot) | apo-exp (blind) | apo-pred (blind) | holo | apo-exp (annot) | apo-pred (annot) | apo-exp (blind) | apo-pred (blind) |
| **DUD-E (n = 38)** | | | | | | | | | | |
| Glide-SP | 0.2958 | 0.1427 | 0.1761 | – | – | 17.25 | 7.74 | 9.16 | – | – |
| RTMScore | 0.4311 | 0.1918 | 0.2077 | – | – | 26.34 | 11.47 | 11.74 | – | – |
| EquiScore | 0.2479 | 0.1466 | 0.1644 | – | – | 14.46 | 8.44 | 9.89 | – | – |
| TankBind | 0.2886 | 0.2996 | 0.3008 | 0.3074 | 0.2930 | 17.03 | 18.13 | 17.76 | 18.52 | 17.36 |
| DrugCLIP | 0.5157 | 0.3493 | 0.3746 | 0.1926 | 0.1974 | 33.70 | 21.36 | 22.70 | 11.75 | 12.05 |
| AANet | **0.6365** | **0.5866** | **0.6003** | **0.5764** | **0.6232** | **40.85** | **38.03** | **38.46** | **37.19** | **40.85** |
| **LIT-PCBA (n = 12)** | | | | | | | | | | |
| Glide-SP | 0.0565 | 0.0503 | 0.0323 | – | – | 5.05 | 3.06 | 1.42 | – | – |
| RTMScore | 0.0445 | 0.0173 | 0.0205 | – | – | 3.34 | 0.67 | 1.16 | – | – |
| EquiScore | 0.0556 | 0.0200 | 0.0511 | – | – | 4.06 | 1.27 | 3.24 | – | – |
| TankBind | 0.0455 | 0.0491 | 0.0472 | 0.0496 | 0.0430 | 3.47 | 3.21 | 3.14 | 3.42 | 2.90 |
| DrugCLIP | 0.0690 | 0.0554 | 0.0483 | 0.0210 | 0.0155 | 5.96 | 4.51 | 2.85 | 1.54 | 0.88 |
| AANet | **0.0850** | **0.0730** | **0.0805** | **0.0630** | **0.0715** | **7.54** | **5.58** | **6.64** | **3.92** | **5.40** |

**From holo to apo with annotated pocket.**

On the both benchmarks, AANet achieves the highest scores in BEDROC and EF1% across *holo* and both *apo* settings, and ranks best in AUROC as well (details in Appendix C.2). Notably, its performance remains steady across structural conditions, demonstrating superior robustness to pocket uncertainty and backbone variation. In contrast, both docking-based methods (including rescoring) and the docking-free DrugCLIP show substantial performance degradation. TankBind maintains consistent performance likely due to its use of detected pockets during training rather than ligand-defined ones, as well as its large binding region radius (up to 20 Å), which in some cases covers the entire protein when the target is small.

**From annotated to blind.**

To evaluate model performance in a fully *apo* scenario—where only *apo* structures are available and no reliable binding pocket annotations are assumed—we provide the model with all cavities detected from the structure and allow it to infer the binding site autonomously. Since a protein may contain multiple potential pockets, but each ligand typically binds to a specific site, we define the final

score for each compound as the maximum score across all candidate pockets associated with that protein. As shown in Table 1, AANet maintains stable performance across metrics, while DrugCLIP continues to decline. Due to the computational burden of the docking search phase under blind settings—whether across multiple candidate pockets or via global search—docking and rescoring methods are not evaluated in this setting. It is unsurprising that TankBind maintains consistent performance, as this phase primarily relies on pocket identification, and its training set was not deduplicated against the targets in either benchmark. Our further analysis on pocket identification reveals that its generalization ability is significantly inferior to that of AANet.

### 4.3 Pocket identification with holo ligand

Table 2: Pocket identification performance on COACH420 dataset (**n = 433** ligands/pockets on 288 structures) at various distance cutoff thresholds (Å) from pocket center to any ligand heavy atom (DCA). "Top-1" is the fraction of cases where the highest-scoring pocket lies within the given DCA; "Top-n" is the fraction where any of the top $n$ pockets lies within that DCA. The oracle row is shaded in gray; bold entries denote the best among non-oracle methods.

| Method | Top-1 (DCA $\leq x$ Å) | | | | Top-n (DCA $\leq x$ Å) | | | |
|---|---|---|---|---|---|---|---|---|
| | 1 | 2 | 3 | 4 | 1 | 2 | 3 | 4 |
| Ideal (Oracle) | 0.1488 | 0.5512 | 0.7070 | 0.7558 | 0.1488 | 0.5512 | 0.7070 | 0.7558 |
| PRANK | – | – | – | – | **0.1290** | 0.3690 | 0.4800 | 0.5440 |
| TankBind | 0.0728 | 0.2887 | 0.3850 | 0.4178 | 0.0798 | 0.3122 | 0.4155 | 0.4507 |
| DrugCLIP | 0.0535 | 0.2651 | 0.3372 | 0.3442 | 0.0605 | 0.2837 | 0.3605 | 0.3721 |
| AANet (w/o agg) | 0.1047 | 0.3837 | 0.5000 | 0.5256 | 0.1140 | 0.4116 | 0.5395 | 0.5698 |
| AANet | **0.1140** | **0.4140** | **0.5419** | **0.5721** | 0.1196 | **0.4349** | **0.5744** | **0.6047** |

Since AANet has demonstrated near-holo performance even under uncertainty in pocket structure and location, we attribute this robustness to AANet's ability to effectively identify and distinguish true binding pockets. We further compare the pocket identification ability of docking-free deep learning methods against the classical ligand-free PRANK method to decompose its power in blind settings.

To further validate this capability, we evaluate AANet on the COACH420 dataset [28]. As shown in Table 2, AANet significantly outperforms the baseline models TankBind and DrugCLIP across all distance thresholds, and consistently achieves the highest top-1 and top-$n$ pocket identification accuracy among non-oracle methods. It is only marginally outperformed by PRANK in the top-$n$ DCA at 1 Å, a setting in which the oracle performance is notably low. These results confirm that AANet not only achieves strong screening performance but also excels at identifying the correct binding pocket among multiple candidates, underlining its robustness under blind *apo* settings.

### 4.4 Ablation and analysis

#### 4.4.1 Ablation on different modules

We conduct ablation studies on the DUD-E dataset ($n = 38$) to evaluate the individual contributions of each module in our framework. For the alignment phase, we use standard contrastive pocket-molecule learning as the baseline and ablate three key components: cavity augmentation, negative sampling, and pocket enlargement. As shown in Table 3, each component contributes positively to performance, demonstrating its effectiveness.

In the aggregation phase, we assess the impact of adapter design by comparing our attention-based adapter with a simple argmax-based alternative. Results show that the attention-based adapter leads to substantial improvement, while the argmax version performs similarly to the model prior to aggregation—highlighting the importance of learning adaptive attention weights. Finally, incorporating pocket label supervision during adaptation yields an additional performance gain.

#### 4.4.2 Generalization to different pocket detection methods.

Since AANet was primarily trained and evaluated using pockets detected by Fpocket, one potential concern is whether its performance stems from overfitting to Fpocket-specific biases rather than

Table 3: BEDROC scores on DUD-E (n = 38) under different ablation settings. CA = cavity augmentation, NS = negative sampling, EP = enlarged pocket; PA = pocket adapter, PL = pocket label supervision. For PA column, M = argmax pocket adapter, A = cross-attention adapter.

| | Module | | | | | BEDROC | |
|---|---|---|---|---|---|---|---|
| | CA | NS | EP | PA | PL | apo-exp (blind) | apo-pred (blind) |
| **Alignment phase** | | | | | | | |
| contrastive pocket-molecule learning | × | × | × | – | – | 0.0838 | 0.0822 |
| + cavity augmentation | ✓ | × | × | – | – | 0.2007 | 0.2341 |
| + negative sampling | ✓ | ✓ | × | – | – | 0.4016 | 0.4097 |
| + enlarged pocket | ✓ | ✓ | ✓ | – | – | **0.4708** | **0.4585** |
| **Aggregation phase** | | | | | | | |
| w/ argmax adapter | – | – | – | M | × | 0.4639 | 0.4620 |
| w/ attention adapter | – | – | – | A | × | 0.5550 | 0.5996 |
| + pocket label (AANet) | – | – | – | A | ✓ | **0.5764** | **0.6232** |

Table 4: Performance of AANet across different pocket detection methods and structural subsets. apo-pred-t denotes truncated predicted *apo* structures.

| Detector | apo-exp | | | apo-pred | | | apo-pred-t | | |
|---|---|---|---|---|---|---|---|---|---|
| | AUROC | BEDROC | EF1% | AUROC | BEDROC | EF1% | AUROC | BEDROC | EF1% |
| Fpocket | 0.7956 | 0.5147 | 32.95 | 0.8359 | 0.5367 | 34.54 | 0.8348 | 0.5589 | 36.06 |
| PocketFinder | 0.8061 | 0.5059 | 32.22 | 0.7620 | 0.4678 | 30.28 | 0.8161 | 0.5434 | 35.23 |
| SURFNET | 0.7906 | 0.4971 | 32.00 | 0.7572 | 0.4331 | 28.01 | 0.8274 | 0.5507 | 35.93 |
| LIGSITE | 0.7989 | 0.4838 | 30.97 | 0.7669 | 0.4532 | 29.13 | 0.8496 | 0.5816 | 38.10 |

learning features intrinsic to the protein structure. To address this, we evaluated AANet using alternative pocket detection methods [25]. As shown in Table 4, AANet maintains a comparable level of performance on experimental *apo* structures among alternative pocket detection methods, while that on predicted ones shows a slight decline. To further investigate this discrepancy, we re-detected pockets and re-evaluated performance on predicted *apo* manually truncated low-confidence regions near the binding site (denoted as apo-pred-t). AANet remained stable across detectors on apo-pred-t, suggesting that the observed gap on full predicted *apo* structures is more likely attributable to differences in robustness of the detection methods to structural inaccuracies, rather than inherent limitations of the model itself. These results indicate that AANet generalizes well across different pocket sources and does not overly rely on artifacts introduced by a specific detector.

### 4.4.3   t-SNE analysis of embedding consistency under pocket uncertainty

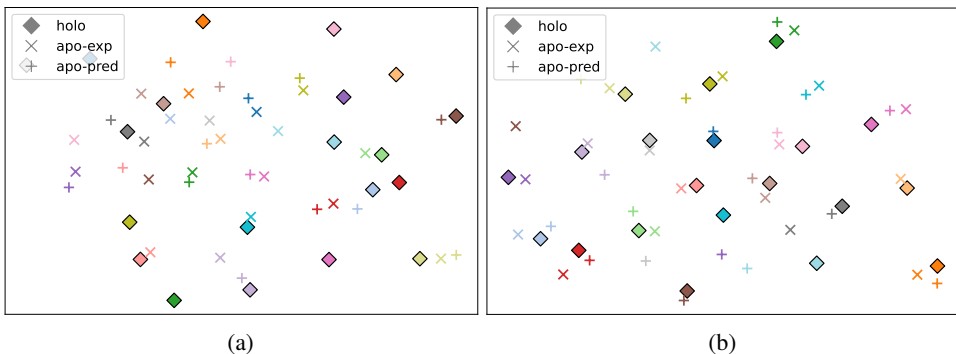

(a)                                                             (b)

Figure 4: t-SNE comparison of pocket–ligand embeddings from three structures. (a) DrugCLIP: embeddings for *holo* and apo-exp/pred (annonated) pockets are widely separated. (b) AANet: embeddings for each target cluster closely.

We randomly selected 20 targets from DUD-E with *holo* and both *apo* pockets (annotated) whose IoU values fell within a moderate range ([0.3, 0.7]). For both DrugCLIP and AANet, pocket and *holo* ligand embeddings were combined via element-wise multiplication to form joint representations, which were projected to two dimensions using t-SNE [35]. As shown in Figure 4, DrugCLIP's embeddings for apo-exp and apo-pred detected pockets lie close to each other but remain distant from the corresponding *holo* embeddings, indicating a lack of robustness to pocket variation. In contrast, AANet produces tightly clustered embeddings across *holo*, apo-exp, and apo-pred conditions for each target, demonstrating strong consistency and pocket-invariant representation.

## 5 Conclusion

We present a new benchmark and systematic evaluation of the challenges faced by DL-based SBVS methods in the *apo* setting, where binding pockets are unknown and structural conformations are often imprecise. To address these challenges, we propose AANet, a novel framework composed of tri-modal alignment and dynamic pocket aggregation. AANet achieves near-holo performance even under blind *apo* conditions. By bridging the gap between AI-based VS and real-world drug discovery needs, our method enables more effective use of predicted structures (e.g., AlphaFold2), extending the applicability of structure-based VS to a wider range of novel protein targets and first-in-class scenarios.

## Acknowledgements

This work is supported by Beijing Academy of Artificial Intelligence and Beijing Frontier Research Center for Biological Structure Fundings.

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

# A Detailed analysis on failure modes

## A.1 Overfitting to the holo pocket

In contrastive pocket–molecule learning (CPML), the pocket is typically defined as any residue within a certain distance (e.g., 6 Å) of ligand atoms during training. This implies an assumption that such ligand-defined pockets are also sufficient for inference. To evaluate how closely detected pockets match the *holo* pocket, we quantify two related but distinct metrics: **coverage**, defined as $Coverage(P_l, P_c) = \frac{|P_l \cap P_c|}{|P_l|}$, and **IoU**, as defined in Equation 3 of the main text. However, as shown in the left side of Figure S1, BEDROC shows little correlation with coverage, but a strong positive correlation with IoU. This indicates that CPML performance depends more on the spatial consistency between the detected and *holo* pockets than on the overall inclusion of ligand-neighboring residues. This suggests an overfitting effect—models may rely heavily on ligand-defined pocket extraction patterns rather than learning generalizable structural representations. In contrast, as shown on the right side of Figure S1, our tri-modal alignment substantially improves early enrichment for targets with moderate IoU values ($0.4 - 0.6$), except for a few with very low IoU ($< 0.3$), where no significant correlation is observed with either coverage or IoU.

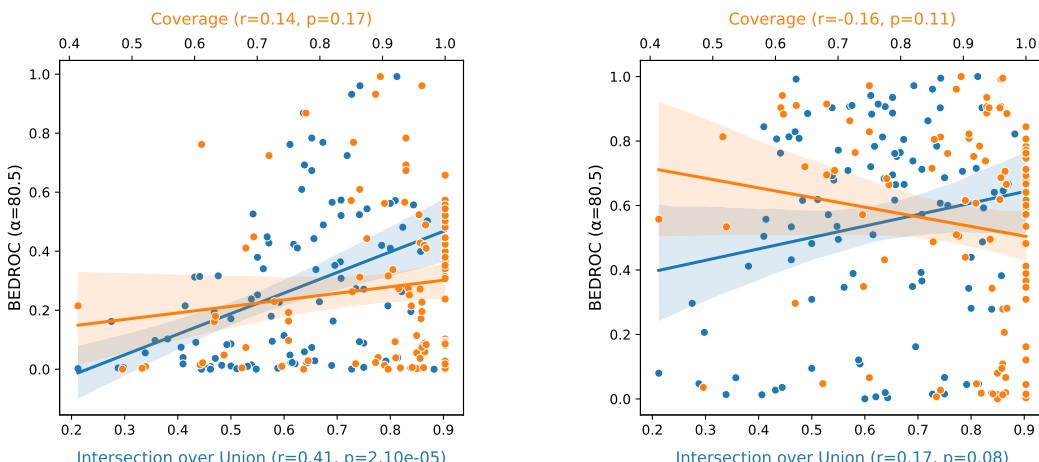

Figure S1: Correlation between BEDROC ($\alpha = 80.5$) and the IoU / coverage of the closest detected pocket to the holo pocket on each target. Left: contrastive pocket–molecule learning; Right: Tri-modal alignment (ours).

## A.2 Low-IoU target cases

In Figure S1, several points exhibit full coverage but lack perfect overlap with the *holo* pocket (i.e., IoU < 1). One such example, HS90A (PDB ID: 1UYG), is visualized on the left side of Figure S2. In this case, the ligand occupies only part of a larger cavity; the detected cavity is a superset of the *holo* pocket. Although this cavity contains sufficient structural information, CPML is disrupted by the additional irrelevant regions—motivating our use of the cavity modality to better capture spatial features. Another representative case, shown on the right side of Figure S2, THRB (PDB ID: 1YPE), involves a cavity that only partially covers the *holo* pocket; the ligand actually spans across two detected cavities. In such scenarios, selecting only one cavity—even if correct—may be insufficient to fully characterize the binding pocket. This observation motivates our pocket enlargement (from 6 Å to 10 Å) strategy.

## A.3 Pocket external shape and ligand shape

Another potential failure mode of ligand-dependent pocket extraction arises from how residues are selected: any residue within a fixed distance of any ligand atom is included, resulting in a coarse, ligand-shaped approximation of the pocket's outer boundary. To illustrate this effect, the ligands from Figure S2 are shown with their corresponding *holo* pocket surfaces in Figure S3.

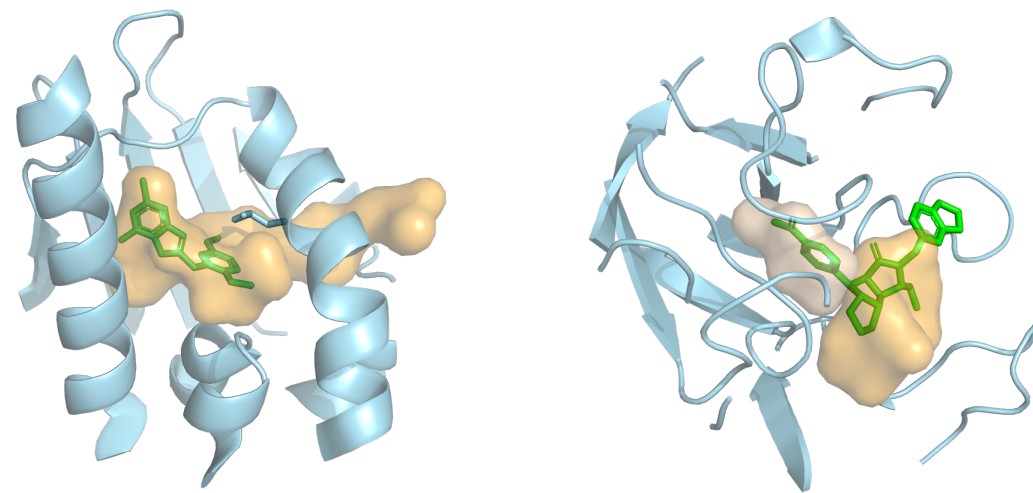

Figure S2: Target cases with low IoU. The protein structure around the ligand is shown in blue, the ligand in green, and the closest detected cavity in orange. **Left**: HS90A (PDB ID: 1UYG), where the detected cavity is a superset of the *holo* pocket. **Right**: THRB (PDB ID: 1YPE), where the detected cavity only partially covers the pocket; an additional cavity corresponding to the remaining pocket region is shown in wheat.

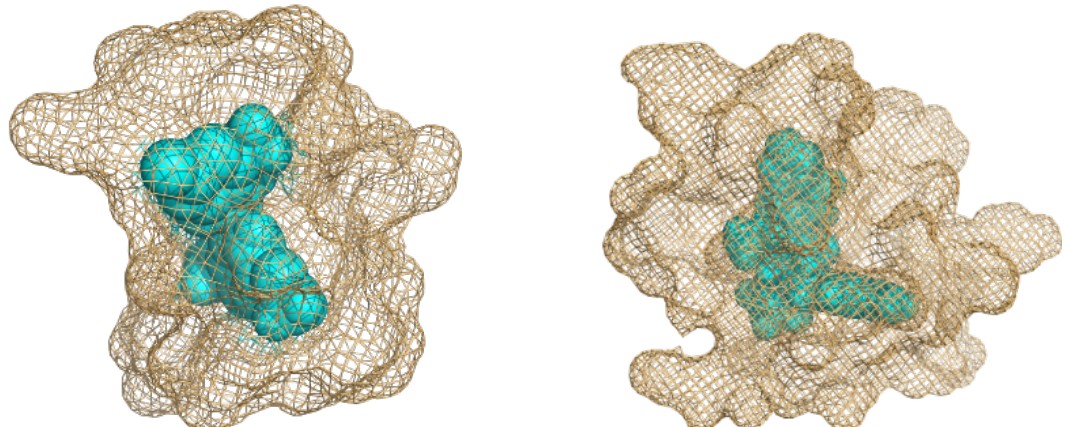

Figure S3: Ligand (atom spheres colored cyan) within the pocket's external surface (rendered as a tan mesh). Left: HS90A (PDB ID: 1UYG); Right: THRB (PDB ID: 1YPE).

Table S1: Correlation between CPML and 3D similarity search methods using the *holo* ligand, along with their enrichment performance. Both Pearson (PRS) and Spearman (SPR) correlations are evaluated on the active set, the full set, and the top 1% of molecules ranked by CPML.

| Similarity | Actives | | Full set | | Top 1% (CPML) | | EF1% | BEDROC $(\alpha = 80.5)$ |
|---|---|---|---|---|---|---|---|---|
| | **PRS** | **SPR** | **PRS** | **SPR** | **PRS** | **SPR** | | |
| USR | 0.0004 | 0.0024 | 0.0014 | 0.0011 | 0.0103 | 0.0062 | 2.74 | 0.0514 |
| PhaseShape | 0.0075 | 0.0066 | 0.0012 | 0.0008 | 0.0048 | 0.0040 | 8.29 | 0.1384 |

To further examine the shape dependence of CPML, we analyze its correlation with two 3D molecular similarity search methods: Ultrafast Shape Recognition (USR) [36], based on atom-distance descriptors, and PhaseShape [37], based on pairwise volume overlap. As shown in Table S1, CPML scores show no meaningful correlation with either method across the active set, the full set, or the top 1% of molecules ranked by CPML. This suggests that CPML may not account for the external pocket shape in enriching molecules with similar 3D shape.

# B Implementation details

## B.1 Pseudo code

---

**Algorithm 1** Cavity Extraction

---

**Require:** protein structure coordinates $x_n$, co-crystallized ligand coordinates $x_m$

    $\{\mathcal{C}^{(s)}\}_{s=1}^S \leftarrow \text{detector}(x_n)$          ▷ each cavity: a set of alpha-sphere centers

    **for** $s = 1$ to $S$ **do**

        $R_c^{(s)} \leftarrow \{\, r \in \text{residues} \mid \exists\, x \in r,\ \exists\, c \in \mathcal{C}^{(s)},\ \|x - c\| \le d \,\}$      ▷ residue-level pocket

    **end for**

**Return:** candidate pockets $\{R_c^{(s)}\}_{s=1}^S$

---

**Algorithm 2** Tri-Modal Contrastive Alignment

---

**Require:** pocket encoder $\mathcal{F}$, cavity encoder $\mathcal{F}_s$, ligand encoder $\mathcal{G}$, candidate cavities $\{P_c^{(s)}\}_{s=1}^S$, holo pocket $P_l$, positive IoU threshold $\tau_{\text{pos}}$, negative IoU threshold $\tau_{\text{neg}}$

    **for** each protein–ligand complex $(x_n, l, P_l)$ **do**

        $IoU(P_l, P_c) = \frac{|P_l \cap P_c|}{|P_l \cup P_c|}$      ▷ compute overlap ratio between holo pocket and candidate

        $P_c \leftarrow \{P_c^{(s)}\}_{s=1}^S$ with $IoU(P_l, P_c) \ge \tau_{\text{pos}}$      ▷ sample one positive cavity

        $\{P_c^-\} \leftarrow$ random half of $\{P_c^{(s)} : IoU(P_l, P_c^{(s)}) \le \tau_{\text{neg}}\}$      ▷ sample hard negatives

        $\mathcal{L}_{\text{CL}} = \mathcal{L}_{p,l}\big(\mathcal{F}(P_l), \mathcal{G}(l)\big) + \mathcal{L}_{p,l}\big(\mathcal{F}_s(P_c), \mathcal{G}(l)\big) + \mathcal{L}_{p,p}\big(\mathcal{F}_s(P_c), \mathcal{F}(P_l)\big)$

        Update parameters of $\mathcal{F}, \mathcal{F}_s, \mathcal{G}$ by minimizing $\mathcal{L}_{\text{CL}}$

    **end for**

**Return:** pretrained pockt encoder $\mathcal{F}$, pretrained cavity encoder $\mathcal{F}_s$, pretrained ligand encoder $\mathcal{G}$

---

**Algorithm 3** Training Cross-Attention Adapter for Cavity Aggregation

---

**Require:** pretrained cavity encoders $\{\mathcal{F}_s\}_{s=1}^S$, pretrained ligand encoder $\mathcal{G}$, trainable adapter (attention weights and projection layer), temperature $t$, bias $b$, weight $\lambda$

    **for** each example $(l, \{P_c^{(s)}\}_{s=1}^S)$ **do**

        $e_l \leftarrow \mathcal{G}(l)$

        **for** $s = 1, \ldots, S$ **do**

            $e_c^{(s)} \leftarrow \mathcal{F}_s\big(P_c^{(s)}\big)$

            $\ell^{(s)} \leftarrow \langle e_l, e_c^{(s)} \rangle / t$      ▷ compute dot-product logits

        **end for**

        $a^{(s)} \leftarrow \exp(\ell^{(s)}) / \sum_{j=1}^S \exp(\ell^{(j)}) \quad \forall s$      ▷ softmax attention

        $\tilde{e}_c \leftarrow \sum_{s=1}^S a^{(s)} e_c^{(s)}$      ▷ aggregate cavities

        $\tilde{e}_l \leftarrow \text{projection}(e_l)$

        compute target distribution $s = [s_1, \ldots, s_S]$:

        **if** complex sample **then**

            $s_s = 1$ for the true cavity, otherwise 0

        **else if** activity-only sample **then**

            $s_s = $ pretrained AANet cavity scores

        **end if**

        $\mathcal{L}'_{\text{KL}} \leftarrow \sum_{s=1}^S s_s \log\big(\frac{s_s}{a^{(s)}}\big)$      ▷ attention supervision

        $\mathcal{L}'_{\text{CL}} \leftarrow \log\big(1 + \exp\big(-t\, \tilde{e}_c \cdot \tilde{e}_l + b\big)\big)$

        $\mathcal{L}_{\text{agg}} \leftarrow \mathcal{L}'_{\text{CL}} + \lambda\, \mathcal{L}'_{\text{KL}}$

        Update adapter parameters to minimize $\mathcal{L}_{\text{agg}}$

    **end for**

**Return:** trained adapter parameters

---

## B.2 ChEMBL data processing

**Molecule filtering**:

- Removed salts or kept only the largest fragment

- Kept molecules with molecular weight between [100, 800]
- Removed those containing atoms other than [H, C, N, O, F, Cl, Br, I, S, P, B, Se]
- Filtered out molecules with unbranched long chains containing 6 atoms or more.

**Activity filtering**: We retained activity records with:

- Confidence score = 9
- Assay type = functional or binding
- Standard type $\in$ ['Ki', 'IC50', 'Kd', 'EC50', 'ED50', 'AC50', 'XC50']
- Values converted to molar units and filtered to lie within the –log10 range [5, 12]

**Target mapping**: Protein targets were mapped to UniProt IDs and matched with AlphaFold structures. Targets without available predicted structures were discarded.

**Contrastive supervision masking**: Each known protein–ligand activity pair was recorded using its UniProt ID and InChI-key. During training, we masked non-diagonal entries that correspond to active (positive) pairs to ensure that true positive pairs are never treated as negatives, even if they are not aligned in the current batch. This masking strategy ensures correct supervision and avoids misleading the model during contrastive learning.

## B.3 Benchmark details

We summarize the PDB entries used as *apo* structures for LIT-PCBA in Table S2. Experimental *apo* structures and AF2-predicted structures were aligned to each reference *holo* PDB using the `structalign` module from the Schrödinger Suite. The resulting alignment scores and root-mean-square deviations (RMSD) are reported in Table S3 for both types of structures. To reduce manual intervention and enable scalability, we adopted a fully automated pipeline for the *apo* (blind) setting. Notably, we retained only non-redundant, non-homologous protein complexes from the apo structures, which increases the overall difficulty of the task and better reflects realistic large-scale virtual screening scenarios.

Table S2: PDB entries selected as *apo* structures for LIT-PCBA targets. Bold chain IDs indicate the primary chain(s) of interest.

| Target Name | UniProt ID | PDB ID | Selected Chain(s) | Comment |
| --- | --- | --- | --- | --- |
| ADRB2 | P07550 | 9chv | **A**, **B**, **C** | Non-redundant protein complex retained |
| ALDH1 | P00352 | 4wj9 | **A** | |
| ESR1 (ago/ant) | P03372 | 2b23 | **A** | |
| FEN1 | P39748 | 5zod | **A** | |
| GBA | P04062 | 3gxd | **A** | |
| IDH1 | O75874 | 1t0l | **A**, **B** | Ligand in holo (6ADG) binds at the interface of two homologous chains |
| KAT2A | Q92830 | 5trm | **A** | |
| MAPK1 | P28482 | 4iz7 | **A**, **B** | Non-redundant protein complex retained |
| PKM2 | P14618 | 1zjh | **A** | |
| PPARG | P37231 | 1prg | **A** | |
| TP53 | P04637 | 1kzy | **A**, **C** | Non-redundant protein complex retained |

Table S3: Alignment scores and RMSD for AF2 and apo structures across LIT-PCBA targets.

| Target | Ref PDB | AF2 | | Apo | |
| | | Score ↓ | RMSD ↓ | Score ↓ | RMSD ↓ |
|---|---|---|---|---|---|
| ADRB2 | 3p0g | 0.106 | 1.609 | 0.238 | 2.398 |
| | 3pds | 0.165 | 2.032 | 0.314 | 2.793 |
| | 3sn6 | 0.107 | 1.592 | 0.225 | 2.336 |
| | 4lde | 0.203 | 2.127 | 0.224 | 2.322 |
| | 4ldl | 0.261 | 2.540 | 0.428 | 3.261 |
| | 4ldo | 0.210 | 2.279 | 0.408 | 3.142 |
| | 4qkx | 0.232 | 2.400 | 0.362 | 2.998 |
| | 6mxt | 0.201 | 2.218 | 0.323 | 2.744 |
| | *mean* | *0.186* | *2.100* | *0.315* | *2.749* |
| ALDH1 | 4wp7 | 0.004 | 0.302 | 0.000 | 0.090 |
| | 4wpn | 0.004 | 0.301 | 0.001 | 0.126 |
| | 4x4l | 0.005 | 0.329 | 0.002 | 0.220 |
| | 5ac2 | 0.006 | 0.345 | 0.002 | 0.211 |
| | 5l2m | 0.004 | 0.307 | 0.001 | 0.136 |
| | 5l2n | 0.005 | 0.329 | 0.001 | 0.154 |
| | 5l2o | 0.009 | 0.463 | 0.004 | 0.301 |
| | 5tei | 0.005 | 0.346 | 0.002 | 0.232 |
| | *mean* | *0.005* | *0.340* | *0.002* | *0.184* |
| ESR1_ago | 1l2i | 0.018 | 0.670 | 0.018 | 0.665 |
| | 2b1v | 0.018 | 0.666 | 0.010 | 0.495 |
| | 2b1z | 0.016 | 0.628 | 0.009 | 0.481 |
| | 2p15 | 0.041 | 1.009 | 0.020 | 0.706 |
| | 2q70 | 0.119 | 1.721 | 0.063 | 1.249 |
| | 2qr9 | 0.032 | 0.898 | 0.016 | 0.622 |
| | 2qse | 0.028 | 0.842 | 0.027 | 0.779 |
| | 2qzo | 0.025 | 0.793 | 0.022 | 0.689 |
| | 4ivw | 0.045 | 0.947 | 0.007 | 0.412 |
| | 4pps | 0.018 | 0.662 | 0.010 | 0.483 |
| | 5drj | 0.021 | 0.726 | 0.009 | 0.481 |
| | 5du5 | 0.015 | 0.616 | 0.012 | 0.551 |
| | 5due | 0.028 | 0.838 | 0.008 | 0.456 |
| | 5dzi | 0.032 | 0.756 | 0.007 | 0.419 |
| | 5e1c | 0.025 | 0.788 | 0.010 | 0.489 |
| | *mean* | *0.032* | *0.837* | *0.017* | *0.598* |
| ESR1_ant | 1xp1 | 0.086 | 1.419 | 0.076 | 1.374 |
| | 1xqc | 0.088 | 1.425 | 0.092 | 1.511 |
| | 2ayr | 0.100 | 1.546 | 0.104 | 1.615 |
| | 2iog | 0.083 | 1.321 | 0.075 | 1.367 |
| | 2iok | 0.089 | 1.208 | 0.070 | 1.174 |
| | 2ouz | 0.090 | 1.473 | 0.093 | 1.521 |
| | 2pog | 0.050 | 1.059 | 0.043 | 0.859 |
| | 2r6w | 0.047 | 1.045 | 0.033 | 0.857 |
| | 3dt3 | 0.130 | 1.783 | 0.101 | 1.590 |
| | 5aau | 0.084 | 1.452 | 0.080 | 1.412 |
| | 5fqv | 0.098 | 1.550 | 0.074 | 1.361 |
| | 5t92 | 0.008 | 0.440 | 0.007 | 0.405 |
| | 5ufx | 0.095 | 1.528 | 0.083 | 1.338 |
| | 6b0f | 0.062 | 1.123 | 0.041 | 1.014 |
| | 6chw | 0.106 | 1.567 | 0.103 | 1.607 |
| | *mean* | *0.081* | *1.329* | *0.072* | *1.267* |
| FEN1 | 5fv7 | 0.056 | 1.181 | 0.038 | 0.970 |
| GBA | 2v3d | 0.014 | 0.549 | 0.026 | 0.778 |
| | 2v3e | 0.012 | 0.510 | 0.027 | 0.752 |
| | 2xwd | 0.006 | 0.376 | 0.028 | 0.727 |
| | 2xwe | 0.013 | 0.557 | 0.030 | 0.770 |

| Target | Ref PDB | AF2 | | Apo | |
|--------|---------|-----------|-----------|-----------|-----------|
| | | Score ↓ | RMSD ↓ | Score ↓ | RMSD ↓ |
| | 3rik | 0.016 | 0.567 | 0.017 | 0.615 |
| | 3ril | 0.017 | 0.599 | 0.011 | 0.468 |
| | *mean* | *0.013* | *0.526* | *0.023* | *0.685* |
| IDH1 | 4i3k | 0.192 | 2.180 | 0.409 | 3.184 |
| | 4i3l | 0.233 | 2.378 | 0.318 | 2.759 |
| | 4umx | 0.209 | 2.238 | 0.310 | 2.746 |
| | 4xrx | 0.172 | 2.068 | 0.388 | 3.107 |
| | 4xs3 | 0.190 | 2.128 | 0.352 | 2.926 |
| | 5de1 | 0.165 | 2.017 | 0.367 | 2.876 |
| | 5l57 | 0.209 | 2.276 | 0.397 | 3.127 |
| | 5l58 | 0.155 | 1.954 | 0.368 | 3.028 |
| | 5lge | 0.103 | 1.589 | 0.258 | 2.534 |
| | 5sun | 0.189 | 2.117 | 0.320 | 2.787 |
| | 5svf | 0.274 | 2.602 | 0.334 | 2.844 |
| | 5tqh | 0.125 | 1.757 | 0.304 | 2.702 |
| | 6adg | 0.179 | 2.074 | 0.372 | 2.987 |
| | 6b0z | 0.193 | 2.143 | 0.366 | 2.979 |
| | *mean* | *0.185* | *2.109* | *0.347* | *2.899* |
| KAT2A | 5h84 | 0.052 | 1.137 | 0.022 | 0.736 |
| | 5h86 | 0.085 | 1.458 | 0.045 | 1.061 |
| | 5mlj | 0.016 | 0.637 | 0.338 | 2.782 |
| | *mean* | *0.051* | *1.077* | *0.135* | *1.526* |
| MAPK1 | 1pme | 0.069 | 1.307 | 0.040 | 0.990 |
| | 2ojg | 0.063 | 1.250 | 0.040 | 1.005 |
| | 3sa0 | 0.082 | 1.424 | 0.047 | 1.085 |
| | 3w55 | 0.074 | 1.358 | 0.101 | 1.590 |
| | 4qp3 | 0.055 | 1.169 | 0.038 | 0.969 |
| | 4qp4 | 0.046 | 1.069 | 0.071 | 1.329 |
| | 4qp9 | 0.051 | 1.132 | 0.053 | 1.145 |
| | 4qta | 0.030 | 0.859 | 0.055 | 1.177 |
| | 4qte | 0.077 | 1.391 | 0.060 | 1.195 |
| | 4xj0 | 0.077 | 1.390 | 0.061 | 1.235 |
| | 4zzn | 0.056 | 1.187 | 0.036 | 0.949 |
| | 5ax3 | 0.059 | 1.190 | 0.075 | 1.365 |
| | 5buj | 0.065 | 1.271 | 0.109 | 1.648 |
| | 5v62 | 0.063 | 1.246 | 0.088 | 1.470 |
| | 6g9h | 0.057 | 1.189 | 0.033 | 0.902 |
| | *mean* | *0.062* | *1.229* | *0.060* | *1.204* |
| PKM2 | 3gqy | 0.141 | 1.837 | 0.030 | 0.741 |
| | 3gr4 | 0.032 | 0.794 | 0.197 | 2.161 |
| | 3h6o | 0.175 | 2.052 | 0.038 | 0.795 |
| | 3me3 | 0.028 | 0.758 | 0.186 | 2.098 |
| | 3u2z | 0.148 | 1.759 | 0.035 | 0.778 |
| | 4g1n | 0.025 | 0.776 | 0.127 | 1.773 |
| | 4jpg | 0.031 | 0.689 | 0.038 | 0.786 |
| | 5x1v | 0.032 | 0.724 | 0.045 | 0.829 |
| | 5x1w | 0.029 | 0.679 | 0.038 | 0.847 |
| | *mean* | *0.071* | *1.119* | *0.082* | *1.201* |
| PPARG | 1zgy | 0.056 | 1.163 | 0.073 | 1.349 |
| | 2i4j | 0.044 | 1.047 | 0.009 | 0.479 |
| | 2p4y | 0.017 | 0.644 | 0.018 | 0.671 |
| | 2q5s | 0.019 | 0.695 | 0.038 | 0.944 |
| | 2yfe | 0.026 | 0.808 | 0.023 | 0.764 |
| | 3b1m | 0.021 | 0.724 | 0.035 | 0.937 |
| | 3hod | 0.045 | 1.058 | 0.007 | 0.415 |
| | 3r8a | 0.025 | 0.797 | 0.032 | 0.893 |

| Target | Ref PDB | AF2 | | Apo | |
|---|---|---|---|---|---|
| | | Score ↓ | RMSD ↓ | Score ↓ | RMSD ↓ |
| | 4ci5 | 0.042 | 0.947 | 0.026 | 0.801 |
| | 4fgy | 0.038 | 0.967 | 0.061 | 1.211 |
| | 4prg | 0.068 | 1.298 | 0.030 | 0.853 |
| | 5tto | 0.018 | 0.665 | 0.017 | 0.659 |
| | 5two | 0.021 | 0.731 | 0.091 | 1.509 |
| | 5y2t | 0.026 | 0.810 | 0.066 | 1.284 |
| | 5z5s | 0.022 | 0.738 | 0.049 | 1.072 |
| | *mean* | *0.033* | *0.873* | *0.038* | *0.923* |
| TP53 | 2vuk | 0.010 | 0.508 | 0.014 | 0.582 |
| | 3zme | 0.010 | 0.498 | 0.015 | 0.607 |
| | 4ago | 0.008 | 0.440 | 0.011 | 0.521 |
| | 4agq | 0.008 | 0.442 | 0.011 | 0.520 |
| | 5g4o | 0.008 | 0.458 | 0.014 | 0.585 |
| | 5o1i | 0.006 | 0.389 | 0.011 | 0.534 |
| | *mean* | *0.008* | *0.456* | *0.013* | *0.558* |
| *overall mean* | | *0.082* | *1.199* | *0.101* | *1.285* |

For details on the DUD-E benchmark, see the supplementary information of [38][3] and [9][4].

### B.4 Hyperparameters

Table S4 summarizes the full set of hyperparameters used to reproduce the results of AANet.

Table S4: Training and evaluation hyperparameters used in this study.

| **Data processing** | |
|---|---|
| Max. No. ligand conformers | 10 |
| Min. RMSD among ligand conformers | 1 |
| Pocket radius | 10 |
| IoU for positives | 0.5 |
| IoU for negatives | 0.1 |
| **Training** | |
| Learning rate | 0.001 |
| Batch size | 48 |
| Random seed | 1 |
| Model hyperparameters | Following DrugCLIP |
| Cavity negative ratio | 0.5 |
| Max. epochs | 200 |
| Early stopping | 10 / 5 |
| Loss logit scale | $\log(10)$ |
| Loss logit bias | $-10$ |
| Adapter softmax temperature | 5 |
| Use fp16 | True |
| **Testing** | |
| Max. No. conformers | 1 |
| Results from multiple pockets (LIT-PCBA) | Max |

[3] https://pubs.acs.org/doi/suppl/10.1021/acs.jcim.0c01354/suppl_file/ci0c01354_si_001.pdf

[4] https://pubs.acs.org/doi/suppl/10.1021/acs.jcim.2c01219/suppl_file/ci2c01219_si_001.pdf

## B.5 Experiments compute resources

To provide a comprehensive estimation of computational resources, we report the main training and inference costs in Table S5. The benchmark evaluations include a combination of targets from DUD-E and LIT-PCBA under multiple structural conditions, including holo, apo, and AF2-derived proteins, as well as different pocket definitions: oracle, annotated, and blind. Each structural condition requires repeated docking and evaluation runs, substantially increasing the cumulative compute burden. In addition to the experiments reported in the main paper, we also performed a number of pilot and ablation studies during model development that are not included in the final results but contributed to the overall resource consumption.

Table S5: Compute resource disclosure.

| Experiment | Resource | Run time | Unit |
|---|---|---|---|
| Training – Alignment phase | $4 \times$ NVIDIA A100 (80 GB) | 2 h | per run |
| Training – Aggregation phase | $4 \times$ NVIDIA A100 (80 GB) | 6 h | per run |
| Model testing (single benchmark) | $1 \times$ NVIDIA A100 (80 GB) | 1–5 min | per benchmark |
| Docking | 128-core CPU server | 3–7 days | per benchmark |
| Baseline evaluation (per DL model) | $1 \times$ NVIDIA A100 (80 GB) | minutes–1 day | per benchmark |

## B.6 Metrics

We evaluate virtual screening performance using three complementary measures that capture both overall ranking quality and early-retrieval effectiveness.

**BEDROC** applies an exponential weight to ranks so that top-ranked actives contribute most to the score. Let $R_i$ be the 1-based rank of the $i$-th active among $N$ compounds, and let $R_a = N_{\mathrm{act}}/N$ be the active fraction. Define

$$Z_\alpha = R_a \frac{1 - e^{-\alpha}}{e^{\alpha/N} - 1}. \tag{S1}$$

Then

$$\mathrm{BEDROC}_\alpha = \frac{\sum_{i=1}^{N_{\mathrm{act}}} e^{-\alpha R_i/N}}{Z_\alpha} \frac{R_a \sinh(\alpha/2)}{\cosh(\alpha/2) - \cosh(\alpha/2 - \alpha R_a)} + \frac{1}{1 - e^{\alpha(1-R_a)}}. \tag{S2}$$

In our experiments we set $\alpha = 80.5$.

**Enrichment Factor (EF$_{\delta\%}$)** quantifies how many more actives are found in the top $\delta\%$ of the ranked list compared to random selection. If $N$ is the total library size, $N_{\mathrm{act}}$ the total number of actives, and $n_{\delta\%}$ the number of actives among the top $k = \lceil \delta N/100 \rceil$ compounds, then

$$k = \left\lceil \frac{\delta N}{100} \right\rceil, \quad \mathrm{EF}_{\delta\%} = \frac{\frac{n_{\delta\%}}{k}}{\frac{N_{\mathrm{act}}}{N}} = \frac{n_{\delta\%} N}{k N_{\mathrm{act}}}. \tag{S3}$$

**AUROC** measures the probability that a randomly chosen active is scored higher than a randomly chosen decoy. Let $\mathrm{TP}(t)$ and $\mathrm{FP}(t)$ be the counts of actives and decoys above score threshold $t$, with $N_{\mathrm{act}}$ and $N_{\mathrm{dec}}$ their totals. Then

$$\mathrm{TPR}(t) = \frac{\mathrm{TP}(t)}{N_{\mathrm{act}}}, \quad \mathrm{FPR}(t) = \frac{\mathrm{FP}(t)}{N_{\mathrm{dec}}}, \tag{S4}$$

and

$$\mathrm{AUROC} = \int_0^1 \mathrm{TPR}\big(\mathrm{FPR}^{-1}(u)\big) \, \mathrm{d}u. \tag{S5}$$

All metrics are computed independently for each target and then averaged over the benchmark.

## B.7 Docking on LIT-PCBA

Due to the high computational cost of docking on LIT-PCBA, we followed PLANET [39] and performed docking using a single *holo* structure, as shown in Table S6, with experimental *apo* and AF2-predicted structures aligned accordingly.

Table S6: PDB codes of protein structures used in the LIT-PCBA virtual screening dataset. Asterisks (*) indicate targets with no available *apo* structure.

| Target Name | PDB ID |
|---|---|
| ADRB2 | 4LDE |
| ALDH1 | 5L2N |
| ESR1_agonist | 2QZO |
| ESR1_antagonist | 5UFX |
| GBA | 2V3D |
| FEN1 | 5FV7 |
| IDH1 | 4UMX |
| KAT2A | 5MLJ |
| MAPK1 | 4ZZN |
| MTORC1* | 4DRI |
| OPRK1* | 6B73 |
| PKM2 | 3GR4 |
| PPARG | 3B1M |
| TP53 | 3ZME |
| VDR | 3A2J |

## B.8 Baseline implementation details

We compare against representative baselines from two categories.

### Docking & Rescoring

**Glide (SP)** [1]: Targets were prepared using the `prepwizard` module with default settings. Molecules were processed using the `ligprep` module to generate up to 32 tautomers and stereoisomers. Docking was performed using the `glide` module with a grid box radius of 10 Å, centered either at the co-crystallized ligand or the closest detected cavity center (in the annotated setting), using Standard Precision mode and default parameters. All modules were from the Schrödinger Suite 2024-1 distribution. The top-ranked conformer for each molecule was retained and subsequently used for rescoring baselines.

**RTMScore** [22]: Implementation obtained from the official GitHub `https://github.com/sc8668/RTMScore` and their pretrained weights on the PDBbind-v2020 [31] general set. The authors excluded any complexes overlapping with the PDBbind-v2020 core set and the CASF-2016 [40] benchmark (reducing from 19,443 to 19,149 complexes), but did not apply filtering against DUD-E or LIT-PCBA. Following the repository defaults, pockets are defined as all receptor atoms within 10 Å of the ligand, or the cavity in the annotated setting, along with the default scoring scheme.

**EquiScore** [21]: We downloaded the code and model weights (trained on PDBscreen, with all proteins from DUD-E and DEKOIS2.0 [41] excluded from the training data, but not those from LIT-PCBA) from `https://github.com/CAODH/EquiScore`. Each molecule's own docking pose was used to define its pocket, i.e., receptor residues within 8 Å. Scores were then obtained using the default inference configuration.

### Docking-free baseline

**TankBind** [23]: We downloaded the official implementation from `https://github.com/luwei0917/TankBind` along with their pretrained weights on PDBbind-v2020 [31]. Following the EquiBind [42] time split, complexes deposited before 2019 (17,787 after RDKit filtering) constituted the training set, those from 2019 (968) the validation set, and those after 2019 (363) the test set, without deduplication against DUD-E or LIT-PCBA. We followed the original settings for all preprocessing and hyperparameters. Pockets were detected using P2Rank [28], with a 20 Å radius

around each predicted center. In the annotated setting, when a true pocket was available, we computed its centroid and selected the single P2Rank pocket whose center was closest. In the blind setting, where no ligand was provided, we scored all P2Rank pockets and retained the maximum predicted affinity per compound.

**DrugCLIP** [3]: We used the official DrugCLIP implementation from `https://github.com/bowen-gao/DrugCLIP`. The model was fine-tuned on the PDBbind-v2019 general set, with all complexes overlapped with those in DUD-E or LIT-PCBA removed to ensure zero-shot evaluation. All other hyperparameters and data processing followed the authors' original settings.

## C  Additional results

### C.1  Benchmarking computational cost on the LIT-PCBA dataset

As our dual-tower backbone does not involve fusion prior to the adapter and supports embedding reuse, AANet introduces minimal overhead beyond DrugCLIP while providing significantly improved performance, particularly in unbound settings. As shown in Table S7, the majority of runtime during the scoring phase arises from data loading and CPU–GPU communication, rather than model inference itself. On typical commercial-scale libraries, AANet maintains ultra-fast screening throughput comparable to DrugCLIP.

Table S7: Comparison of computational cost across methods. *Sharing cost refers to preprocessing that can be reused. **PCBA-One corresponds to the subset of 2.8M complexes.

| Method | Sharing Cost* | PCBA-One** (2.8M) | PCBA-Full (10M) | PCBA-Apo-Blind (100M) | Overall Cost |
|---|---|---|---|---|---|
| Glide | LigPrep: 1 hr (930 mols/s) | 3 days | 11 days | 3.6 months | 3 days – 3.6 months |
| RTMScore | – | Preprocessing: 3 hrs (247/s) Inference: 8.2 hrs (94/s) | 11.2 hrs + 29.5 hrs | 17 days | 11.2 hrs – 17 days |
| EquiScore | – | Preprocessing: 2.7 hrs (291/s) Inference: 8.1 hrs (96/s) | 11 hrs + 29 hrs | 16.7 days | 10.8 hrs – 16.7 days |
| TankBind | – | 7 hrs | 25 hrs | 10.4 days | 7 hrs – 10.4 days |
| DrugCLIP | Mol: 270 s (10,370/s) Pocket: 1–2 s (86/s) | – | 15 s | 15 s | ~300 s |
| AANet | Mol: 270 s (10,370/s) Pocket: 1–3 s (36/s) | – | 15 s | 15 s | ~300 s |

### C.2  AUROC under annotated and blind settings

Table S8 reports the AUROC of all methods on DUD-E and LIT-PCBA across annotated and blind structural settings. While AUROC reflects overall ranking quality, it is less indicative of early retrieval performance, which is more critical for virtual screening given practical experimental costs.

### C.3  Results for holo full sets

We benchmarked AANet and the baselines under conventional *holo* settings, where *holo* complexes were provided and pockets were defined by the co-crystallized ligands, as shown in Table S9.

### C.4  Results on the DEKOIS dataset

we mapped the DEKOIS targets to UniProt ID and successfully downloaded 77 target structures from the AlphaFold Database, excluding 4 virus-related targets that are not available (HIV1PR & HIV1RT: P03366, SARS-HCoV: P0C6X7, NA: P27907).

As shown in Table S10, our method consistently outperforms DrugCLIP across all settings, and surpasses traditional docking baselines under holo pocket settings (*italic*) when tested under the most challenging apo-pred blind setup (**bolded**).

### C.5  Results on the DUD-E dataset under de-redundant / de-homologous setting

On PDBbind, we removed duplicate complexes based on structure IDs, following prior work to ensure fair comparison. However, this may not fully eliminate target-level redundancy. For some of baselines, duplicates were not removed (Appendex B.8). On ChEMBL, we filtered data using UniProt IDs, which more reliably removes overlapping targets.

Table S8: AUROC performance on DUD-E and LIT-PCBA. Each method is evaluated on three structural subsets: holo, apo (experimental), and apo (predicted), under both annotated and blind settings. Row colors indicate method type: Docking & Rescoring , Docking-free baseline , and Proposed method .

| Method | Holo | apo-exp (annot) | apo-pred (annot) | apo-exp (blind) | apo-pred (blind) |
|---|---|---|---|---|---|
| **DUD-E (n = 38)** | | | | | |
| Glide-SP | 0.6054 | 0.5461 | 0.5588 | – | – |
| RTMScore | 0.7211 | 0.6077 | 0.6246 | – | – |
| EquiScore | 0.7257 | 0.6478 | 0.6631 | – | – |
| TankBind | 0.7872 | 0.7765 | 0.8006 | 0.7745 | 0.7897 |
| DrugCLIP | 0.8133 | 0.7518 | 0.7826 | 0.6134 | 0.5798 |
| AANet | 0.8963 | 0.8744 | 0.8752 | 0.8593 | 0.8617 |
| **LIT-PCBA (n = 12)** | | | | | |
| Glide-SP | 0.5078 | 0.5070 | 0.5037 | – | – |
| RTMScore | 0.5245 | 0.5161 | 0.5324 | – | – |
| EquiScore | 0.5800 | 0.4694 | 0.5342 | – | – |
| TankBind | 0.6110 | 0.6043 | 0.6121 | 0.6119 | 0.5942 |
| DrugCLIP | 0.5742 | 0.5851 | 0.5635 | 0.5006 | 0.4805 |
| AANet | 0.5621 | 0.5737 | 0.5640 | 0.5750 | 0.5641 |

Table S9: Performance comparison on DUDE and LIT-PCBA benchmarks under conventional *holo* setting.

| Method | DUDE | | | LIT-PCBA | | |
|---|---|---|---|---|---|---|
| | AUROC | BEDROC ($\alpha = 80.5$) | EF 1% | AUROC | BEDROC ($\alpha = 80.5$) | EF 1% |
| *Docking & Rescoring* | | | | | | |
| Glide-SP | 0.7670 | 0.4070 | 16.18 | 0.5315 | 0.0400 | 3.41 |
| Vina | 0.7160 | – | 7.32 | 0.5693 | 0.0370 | 1.71 |
| NN-score | 0.6830 | 0.1220 | 4.02 | 0.5570 | 0.0250 | 1.70 |
| RFscore | 0.6521 | 0.1241 | 4.52 | 0.5710 | – | 1.67 |
| Pafnucy | 0.6311 | 0.1650 | 3.86 | – | – | 5.32 |
| OnionNet | 0.5971 | 0.0862 | 2.84 | – | – | – |
| RTMScore | 0.7529 | 0.4341 | 27.10 | 0.5247 | 0.0388 | 2.94 |
| EquiScore | 0.7760 | 0.4320 | 17.68 | 0.5678 | 0.0490 | 3.51 |
| *Docking-free* | | | | | | |
| TankBind | 0.7509 | 0.3300 | 13.00 | 0.5970 | 0.0389 | 2.90 |
| Planet | 0.7160 | – | 8.83 | 0.5731 | – | 3.87 |
| DrugCLIP | 0.8093 | 0.5052 | 31.89 | 0.5717 | 0.0623 | 5.51 |
| AANet | 0.8510 | 0.5592 | 36.05 | 0.5353 | 0.0677 | 5.13 |

To further address potential leakage, we conducted an additional experiment by using MMseqs2 [43] to identify and remove any training proteins (from PDBbind and ChEMBL) containing chains with $\geq$ 90% sequence identity to any DUD-E structure / target sequences. We then compared our method and DrugCLIP under both the original and this more stringent de-redundant / de-homologous setting. As shown in Table S11, our method maintains robust performance even after removing homologous sequences, demonstrating better generalization.

## C.6 Comparison with ligand-based methods

ligand-based virtual screening (LBVS) can bypass the need for protein structural information and is particularly useful when high-quality ligands are available or when protein structures are unreliable.

Table S10: Performance on DEKOIS (77 targets) under holo, apo-pred-annotated (the cavity detected by FPocket that is closest to the holo pocket), and apo-pred-blind settings.

| Method | AUROC (holo) ↑ | AUROC (apo-annot) | AUROC (apo-blind) | BEDROC (holo) ↑ | BEDROC (apo-annot) | BEDROC (apo-blind) | EF@0.5% (holo) ↑ | EF@0.5% (apo-annot) | EF@0.5% (apo-blind) |
|---|---|---|---|---|---|---|---|---|---|
| Vina | – | – | – | – | – | – | 2.58 | – | – |
| GOLD | – | – | – | – | – | – | 10.32 | – | – |
| Glide | – | – | – | – | – | – | *11.85* | – | – |
| DrugCLIP | 0.7967 | 0.7075 | 0.4355 | 0.532 | 0.322 | 0.0313 | 20.62 | 12.12 | 0.92 |
| AANet | 0.8785 | 0.8624 | 0.7708 | 0.6446 | 0.5824 | 0.3891 | 23.26 | 20.91 | **13.33** |

Table S11: Performance on DUD-E under de-redundant and de-homologous (90%) settings.

| Method | BEDROC (holo) | BEDROC (apo-pred annot) | BEDROC (apo-exp annot) | BEDROC (apo-pred blind) | BEDROC (apo-exp blind) | EF@1% (holo) | EF@1% (apo-pred annot) | EF@1% (apo-exp annot) | EF@1% (apo-pred blind) | EF@1% (apo-exp blind) |
|---|---|---|---|---|---|---|---|---|---|---|
| DrugCLIP (de-redundant) | 0.5157 | 0.3746 | 0.3493 | 0.1974 | 0.1926 | 33.70 | 22.70 | 21.36 | 12.05 | 11.75 |
| DrugCLIP (de-homologous) | 0.3949 | 0.2323 | 0.2134 | 0.1097 | 0.1298 | 24.90 | 13.56 | 12.59 | 6.51 | 7.36 |
| AANet (de-redundant) | **0.6365** | **0.6003** | **0.5866** | **0.6232** | **0.5764** | **40.85** | **38.46** | **38.03** | **40.85** | **37.19** |
| AANet (de-homologous) | *0.5985* | *0.5247* | *0.5475* | *0.5955* | *0.5411* | *38.55* | *33.59* | *34.76* | *38.34* | *34.50* |

However, LBVS is often more straightforward when good query ligands are available (e.g., potent, exogenous ligands), but may be less effective in identifying novel chemotypes or in the absence of relevant prior ligands. In contrast, SBVS leverages the protein's structural context, which can help discover structurally diverse actives but depends on structure quality.

We evaluated several widely-used LBVS methods, covering both 2D and 3D paradigms, including ECFP (2D topology-based), USR (3D shape-based), and Shape from Schrödinger suite (3D shape overlap). The results are summarized in Table S12.

Table S12: Performance of common LBVS methods on DUD-E benchmark.

| Method | EF@1% | BEDROC ($\alpha = 80.5$) |
|---|---|---|
| USR [36] | 2.74 | 0.051 |
| PhaseShape [37] | 8.29 | 0.138 |
| ECFP4 [44] | 26.61 | 0.414 |
| AANet (apo-blind) | 37.19 | 0.576 |

The notably strong performance of ECFP4 is likely due to the fact that many actives in virtual screening benchmarks such as DUD-E share similar topological scaffolds, whereas decoys are specifically designed to have similar physicochemical properties but divergent topologies. As a result, ECFP4 can effectively distinguish actives from decoys based on topological similarity alone, leading to higher enrichment factors. However, such advantages may not always generalize to real-world screening scenarios, where novel actives are often needed.

## D    Additional Discussion

### D.1    Selection of the pocket extraction radius

This choice of 10 Å pocket extraction radius was based on quantitative analysis on the PDBbind dataset, supported by the following three key findings:

1. **Coverage of original holo pockets**: The oracle 10 Å cavities, which are defined as those with the highest overlap (IoU) with holo pockets, fully cover 75.38% of the original 6 Å holo pockets, and 90.94% achieve $\geq 85\%$ coverage. This indicates that 10 Å cavities provide a reliable approximation of the ground-truth pocket regions.

2. **Tradeoff between feature integrity and computational cost**: The average atom count in 10 Å oracle cavities is $612.68 \pm 240.75$, over 3× larger than the 6 Å holo pockets ($199.32 \pm 55.04$). Since computational cost scales non-linearly with the number of input atoms, we uniformly downsample pockets to $\leq 256$ atoms during training to ensure fair comparisons across different pocket definitions.

Larger regions would require more aggressive downsampling, which may compromise spatial features or increase computational overhead. Thus, 10 Å represents a practical upper bound that balances representational fidelity with efficiency.

3. **Avoiding bias from over-extension**: Extending the radius beyond 10 Å risks incorporating off-target cavities, especially on cluttered protein surfaces. As our 10 Å is defined by minimal atom-to-atom distance (not center-based), it already captures ample structural context. Further expansion would dilute the binding signal and potentially introduce irrelevant patterns.

Thus, the 10 Å threshold strikes a practical balance: it ensures high coverage of the ground-truth binding pocket, preserves structural integrity, and avoids unnecessary computational overhead or learning from irrelevant patterns.

## D.2 The role of holo pocket in tri-modal contrastive learning

The alignment with holo pockets plays a critical and **bidirectional** role in our tri-modal contrastive framework, motivated by the following:

1. **Holo $\rightarrow$ Cavity - Guidance for learning from noisy input**: Holo pockets, shaped by actual ligand binding, serve as reliable supervisory anchors (as stated by the reviewer). Aligning detected cavities with holo pockets helps the model extract meaningful features even from noisy or imprecise cavity detection.

2. **Cavity $\rightarrow$ Holo - Addressing structural uncertainty and generalization**: Holo pockets are biased toward a specific ligand series (Appendix A), while other actives may bind elsewhere. By aligning cavities (which are ligand-agnostic) to holo pockets, the model learns features that are spatially intrinsic to the protein. This improves the model's robustness to structural uncertainty and enhances generalization. This alignment also boosts performance on holo structures, since the learned features are less tied to ligand-specific conformations.

3. **Cavity $\leftrightarrow$ Holo - Dual modeling of general and specific signals**: The model jointly learns ligand-independent (intrinsic to proteins) and ligand-specific pocket representations, which improves robustness. This enables stronger performance across both holo and apo structures, not just mitigating the performance gap but outperforming DrugCLIP in both cases.

In summary, holo-cavity alignment is essential for modeling both precise and generalizable binding patterns, supporting our goal of robust virtual screening under structural uncertainty.

## D.3 Strengths and limitations of pocket detection

While our current framework employs Fpocket for cavity detection during training, AANet demonstrates robustness across different cavity detectors. Although our model uses Fpocket to extract training cavities, the downstream performance remains stable across different cavity detection tools (Table 4). These conventional detectors are grounded in physicochemical definitions of binding pockets, such as surface concavity, void volume, and geometric constraints. The consistency across tools suggests that AANet is not overfitting to a specific detection algorithm, but instead learns underlying structural priors intrinsic to proteins, such as spatial enclosures or clefts.

**Alignment with practical structure-based pipelines**: Our method follows the typical structure-based virtual screening pipeline, where pocket detection and binding assessment are distinct stages. This modularity aligns with standard workflows in drug discovery and allows our method to plug into existing infrastructure while remaining adaptable to alternative cavity detectors.

**Physically meaningful supervision**: By grounding learning on cavities defined via physical heuristics (e.g., void detection), we introduce inductive bias from medicinal chemistry intuition into the model. This helps prevent overfitting to spurious binding patterns and encourages the model to focus on biophysically plausible pockets. Concerns about physical plausibility have recently gained attention in the structure prediction and docking communities, as highlighted by works such as PoseBusters in deep learning-based docking. This emphasis on biophysical realism is equally relevant to virtual screening and motivates the design of our method.

Although we currently do not explore **end-to-end cavity prediction**, which is a promising direction, especially on novel targets including flexible proteins. As datasets grow and modeling techniques evolve, we believe this integration could offer even more robust and generalizable solutions.

**Limitation in flexible or disordered proteins**: We acknowledge that for highly flexible or intrinsically disordered proteins, where well-formed pockets are absent and ligand-induced fit is essential, existing pocket-based SBVS methods, including ours, face inherent challenges. Our work primarily addresses uncertainty in binding site localization, not full flexible binding modeling.

