# OpenReview forum: "AANet: Virtual Screening under Structural Uncertainty via Alignment and Aggregation"
_NeurIPS.cc/2025/Conference — NeurIPS 2025 poster_

### Official Review · Reviewer_x4Dm · 2025-06-30

**Clarity:** 4
**Significance:** 3
**Originality:** 4
**Rating:** 5
**Confidence:** 4

**Summary:**

The authors introduce a contrastive learning-based approach that aims to allow docking with apo and holo structures. They introduce AANet, which uses pockets identified from Alphafold structures and learns alignments with the ground truth holo pockets and experimentally determined apo structures. As a negative loss signal, they use cavities that are not suitable docking candidates. This contrastive learning approach allows their model to exhibit similar performance across apo/holo structures, and they can show that their model learns a similar representation for similar pockets and that their model outperforms current models in almost all areas.

**Questions:**

1. The authors say that they use Fpocket to determine potential pockets and randomly select one candidate cavity. I was wondering if the same pocket is selected in each training step (considering the same setup), or if it changes during the training and a new candidate is chosen each time?

2. Do you think your approach could benefit from more augmentations? E.g., randomized positions, different Alphafold structures (maybe from esmfold, boltz1 etc.)

3. The part where the authors introduce training with pocket-agnostic data is not particularly clear to me and I think the clarity of this part could be improved. I am not sure if first, the model is trained with contrastive loss and then fine-tuned with the pocket-agnostic data, or both of them are optimized simultaneously? Also, I was wondering about the specific choice for $\lambda$.

4. In the manuscript, the authors say that the model is trained "with all entities overlapping with DUD-E or LIT-PCBA removed" to avoid data leakage. Similar lines are found in multiple places. I was wondering, whether they accounted for structure similarity, because it seemed like they were only looking at exact matches for IDs. Without structure similarity, this could still lead to data leakage.

5. In 4.4.2 the authors demonstrate that their model exhibits similar performance across pocket finding algorithms, although we see a slight decline in some cases. I would like to see if training a model on pockets randomly sampled from multiple different approaches could be beneficial and allow the model to better generalize.

**Ethical Concerns:**

["NO or VERY MINOR ethics concerns only"]

**Final Justification:**

I believe that the paper would be a good fit for the conference, as it shows a well-executed idea and the authors demonstrate consistent results throughout their benchnmarks. Also reviewing the author responses, it seems like the reviewers share a similar opinion that the manuscript could be good fit for the conference.

During the rebuttal, the authors have run new experiments and provided additional results that will further strengthen the manuscript.

As a core weakness, I listed the missing standard deviations and error bars. During the rebuttal, the authors have provided some more additional results, but I think this should have been done for all experiments. Still, as the works they benchmark against do not provide any error bars either, I think it is okay in the current form.

Another important aspect for me is the discussion about structure similarity in the benchmark. They have fully addressed this during the rebuttal and would like that they include this table in the revised manuscript.

Overall, I believe that this is a high-quality paper suitable for this conference.

**Limitations:**

yes

**Paper Formatting Concerns:**

No formatting concerns

**Quality:**

4

**Strengths And Weaknesses:**

The manuscript is of high quality, and seems polished, easy to read, and all results are well presented and documented. Their method shows rather consistent improvements across all demonstrated benchmarks. The presented idea to incorporate contrastive learning is not novel per se, but well executed and the specific tri-modal alignment can be beneficial for the community.

The main weakness I see in this paper is that they do not report any standard deviations or error bars, which makes comparison of some benchmarks difficult. Also, the authors could have compared with more methods and I would like to have seen a more detailed comparison with other related methods, for instance in a related work section.

Overall, the authors show a variety of benchmarks and ablation studies, showcasing the limitations and contributions of their approach well.

---

> ### Author Rebuttal · Authors · 2025-07-31
>
> > W1: No standard deviations or error bars.
>
> We did not include standard deviations initially because most prior studies do not report results or offer pre-trained models from multiple runs, making direct comparison of variability infeasible. However, to address this concern, we have conducted four parallel runs of our method on the DUD-E benchmark to estimate variability. The resulting standard deviations are now included in Table 1.
>
> **Table 1.** Performance (mean ± std) on the DUD-E benchmark.
>
> | AANet  | holo  | apo-exp-annotated | apo-pred-annotated | apo-exp-blind  | apo-pred-blind |
> | - | - | - | - | - | - |
> | BEDROC | 0.6324±0.0074 | 0.5737±0.0088 | 0.5912±0.0061  | 0.5706±0.0061 | 0.6099±0.0090 |
> | EF@1%  | 40.73±0.51 | 37.12±0.57 | 37.59±0.58  | 37.00±0.54 | 39.91±0.68 |
>
> We hope this addition improves the clarity and reproducibility of our reported results.
>
> > W2: Broader method comparison and related work discussion.
>
> We thank the reviewer for the suggestion. Since we evaluate our model on newly curated benchmark settings, all baselines had to be **re-implemented or re-run**, which limited the scope of direct comparisons. Some previously published methods used in related studies either lack publicly available inference code or are outdated. We include this comparison with previously reported results in Supplementary Section C.2.
>
> We have further expanded our comparisons to include the following two categories:
>
> * Several representative ligand-based virtual screening (LBVS) methods (ECFP, USR, PhaseShape from the Schrödinger suite):
>
> | Method   | EF@1% | BEDROC (α=80.5) |
> | - | - | - |
> | USR [1]  | 2.74  | 0.051   |
> | PhaseShape [2] | 8.29  | 0.138   |
> | ECFP4 [3]   | 26.61 | 0.414   |
> | AANet (apo-blind) | 37.19 | 0.576   |
>
> [1] J Comput Chem. 2007, 28, 10, 1711-23.
> [2] J. Chem. Inf. Model. 2011, 51, 10, 2455–2466.
> [3] J. Chem. Inf. Model. 2010, 50, 5, 742–754.
>
> * A flexible docking method (FlowDock) based on flow matching.
>
> | Method | AUROC ↑ | BEDROC (α=80.5) ↑ | EF@1% ↑ | EF@5% ↑ |
> | - | - | - | - | - |
> | FlowDock [1] | 0.5863   | 0.2358  | 8.15  | 5.24  |
> | AANet  | 0.8651   | 0.6729  | 24.38 | 12.12 |
>
> [1] FlowDock: Geometric Flow Matching for Generative Protein-Ligand Docking and Affinity Prediction, ISMB 2025.
>
> Due to computational constraints, FlowDock was evaluated on a subset of DUD-E compounds, which may underestimate its EF values (roughly halved). We acknowledge that **flexible docking frameworks like FlowDock are not primarily designed for large-scale virtual screening tasks**. Instead, they are better suited for **post-screening re-ranking or refinement stages**, where a small number of candidate actives are already identified. We will further enrich the related work section to reflect these comparisons and position our method more clearly in the context of existing approaches.
>
> > Q1: Pocket selection consistency during training?
>
> We thank the reviewer for this question. During training, we **randomly sample one candidate cavity on the fly** from either the set of positive (overlapping with holo pocket) or negative (non-overlapping) Fpocket-detected pockets on the same protein structure. If no positive pocket is available, a negative one is used instead.
>
> Importantly, all sampling is controlled by a random seed, ensuring reproducibility: for a given dataset, seed, and training setup, the same pocket will be selected for the same sample at the same epoch across different runs. This design balances stochastic training dynamics with full reproducibility.
>
> > Q2: Could the method benefit from more augmentations (e.g., randomized positions, alternative predicted structures)?
>
> We thank the reviewer for this valuable suggestion. Due to computational constraints, we were unable to explore such augmentations in this work. However, we agree that these strategies are promising, especially for future engineering-scale models.
>
> 1. **Disturbed pocket positions** may serve as an effective regularization strategy. However, fully random centers could undermine the **spatial rationale defined by pocket detection tools**, which often capture protein-intrinsic cavity features.
> 2. Using **diverse predicted structures** is a highly promising augmentation. Recent studies (e.g., shallow MSA-based predictions [1], AF-Cluster [2]) show that AlphaFold can generate **alternative conformations representing distinct functional states** (e.g., DFG-in/out in kinases; transporters and G-protein-coupled receptors). Such sampling could help the model learn flexible pocket features. Similarly, including structures from alternatives such as ESMFold or Boltz1 may improve generalization by exposing the model to **distributional variability** in predicted structures.
>
> We believe these augmentations are valuable actions for improving robustness under structural uncertainty.
>
> [1] Elife, 2022, 11, e75751.
> [2] Nature, 2024, 625(7996), 832–839.
>
> > Q3: Clarification on training with pocket-agnostic data: staged vs. joint optimization? How is λ chosen?
>
> We thank the reviewer for pointing this out and will improve the clarity in the final version.
>
> Our training follows a **two-stage design**:
>
> 1. **Stage 1 (pocket-aware)**: We train a dual-tower model with contrastive loss on paired pocket–ligand data.
> 2. **Stage 2 (pocket-agnostic)**: We freeze the backbone encoders and introduce lightweight adapters—a cross-attention adapter on the pocket side (approximating an argmax match) and an identity-like linear adapter on the ligand side. This preserves the model’s ability to identify binding pockets from the original contrastive alignment, while enabling it to effectively leverage pocket-agnostic data for further learning.
>
> This design allows the model to reuse DrugCLIP-style embeddings and enables efficient inference while enhancing learning from broader activity/binding datasets.
>
> As for the **λ setting**, we use a simple value of **1.0**. At this scale, the contrastive loss is naturally two orders of magnitude larger than the KL loss, providing sufficient regularization without dominating training. A larger λ overly constrains the model (resulting in little learning), while a smaller λ reduces its regularizing effect.
>
> > Q4: Did the authors account for structural similarity when removing overlapping entities with DUD-E or LIT-PCBA to prevent data leakage?
>
> We thank the reviewer for this important question. We acknowledge that **ID-based filtering does not eliminate structurally similar entries**. We followed this approach primarily to **align with prior work and ensure fair comparisons**. Notably, some baselines did not apply such filtering at all (cf. Supplementary Section B.7).
>
> To further mitigate potential leakage, we performed an additional analysis using MMseqs2 to identify and exclude any training proteins (from PDBbind and ChEMBL) that contain chains with **≥90% sequence identity** to any DUD-E structure or target sequence. We then compared our method and DrugCLIP under both the original de-redundant and this **stricter de-homologous setting**.
> As shown in Table 4, our method maintains strong performance even after removing homologous sequences, demonstrating superior generalization.
>
> **Table 4.** Performance on DUD-E under de-redundant and de-homologous (90%) settings.
>
> | Method | BEDROC (holo) | BEDROC (apo-pred annot) | BEDROC (apo-exp annot) | BEDROC (apo-pred blind) | BEDROC (apo-exp blind) | EF@1% (holo) | EF@1% (apo-pred annot) | EF@1% (apo-exp annot) | EF@1% (apo-pred blind) | EF@1% (apo-exp blind) |
> | - | - | - | - | - | - | - | - | - | - | - |
> | DrugCLIP (de-redundant)  | 0.5157  | 0.3746   | 0.3493  | 0.1974   | 0.1926  | 33.70  | 22.70   | 21.36  | 12.05   | 11.75  |
> | DrugCLIP (de-homologous) | 0.3949  | 0.2323   | 0.2134  | 0.1097   | 0.1298  | 24.90  | 13.56   | 12.59  | 6.51 | 7.36   |
> | AANet (de-redundant)  | **0.6365** | **0.6003**  | **0.5866** | **0.6232**  | **0.5764** | **40.85** | **38.46**  | **38.03** | **40.85**  | **37.19** |
> | AANet (de-homologous) | *0.5985* | *0.5247*  | *0.5475* | *0.5955*  | *0.5411* | *38.55* | *33.59*  | *34.76* | *38.34*  | *34.50* |
>
> > Q5: Could training a model on pockets randomly sampled from multiple detection methods improve generalization?
>
> We thank the reviewer for this suggestion. We investigated this possibility and found the following:
>
> 1. The **slight performance drop** observed with certain pocket detection tools largely stems from **individual tool failures on specific targets**, especially on AF2-predicted structures. For example, PocketFinder failed to detect the correct pocket for IGF1R, PTPN1, and RENI (IoU < 0.1); SurfNet failed on EGFR, IGF1R, RENI, and RXRA. In contrast, Fpocket consistently detected valid pockets, making it more robust overall.
> 2. In early-stage experiments (Stage 1 model only, without adapters), we augmented training data using **positive pockets from three other detection tools**. This yielded no consistent improvement: for example, EF@1% on the AF2-annotated setting dropped slightly (34.4 → 33.7), while AF2-blind improved modestly (29.3 → 30.7). Similar minor improvements were observed across apo settings. Given AANet’s demonstrated robustness across tools, we believe the model has already internalized spatial priors shared across detection algorithms, reducing the benefit of this augmentation strategy.

---

> > ### Comment · Reviewer_x4Dm · 2025-08-04
> >
> > I thank the authors for responding to all my points in detail. I believe that they have answered my questions fully and I am thankful for the authors reporting some standard deviations.
> >
> > I still think that my initial assessment accurately reflects the state of the manuscript and will leave my score unchanged.

---

### Official Review · Reviewer_TXmb · 2025-07-01

**Clarity:** 3
**Significance:** 3
**Originality:** 3
**Rating:** 5
**Confidence:** 4

**Summary:**

The paper presents AANet, a contrastive learning-based virtual screening framework designed to handle structural uncertainty in protein structures, particularly apo or AlphaFold2-predicted conformations where pocket annotations are absent or unreliable. AANet consists of two main components: (1) tri-modal alignment between ligands, holo pockets, and geometry-derived cavities; and (2) cross-attention-based aggregation to dynamically infer binding pockets from candidate cavities. The method is evaluated on curated DUD-E and LIT-PCBA subsets in both annotated and blind apo settings. It significantly outperforms strong baselines such as DrugCLIP, TankBind, and Glide, and demonstrates robustness across different pocket detection tools.

**Questions:**

1. Is there any discussion of ligand-based virtual screening methods (LBVS)? For example, traditional LBVS methods [1] and recent deep learning-based approaches [2] can bypass the need for protein structural information entirely.

2. Is removing overlapping UniProt IDs sufficient to prevent information leakage between training and test sets? Given that homologous proteins with very high sequence similarity (e.g., >90%) may still exist in training sets, how do the authors ensure that models are not indirectly exposed to structurally or functionally similar targets?

[1] Liu X, Jiang H, Li H. SHAFTS: a hybrid approach for 3D molecular similarity calculation

[2] Zhou G, Wang Z, Yu F, et al. S-MolSearch: 3D Semi-supervised Contrastive Learning for Bioactive Molecule Search

**Ethical Concerns:**

["NO or VERY MINOR ethics concerns only"]

**Final Justification:**

The authors have clarified the key concerns I raised, especially regarding dataset preprocessing, generalization across homologs, and LBVS comparisons. Based on the clarifications and new results, I am raising my score.

**Limitations:**

1. The model relies on external cavity detection (e.g., Fpocket). If the cavity predictions are poor—especially in complex or flexible proteins—the model’s performance may degrade.
2. AANet does not estimate confidence or uncertainty in its predictions, even under structurally noisy or blind settings, which can hinder decision-making in practical pipelines.
3. The data split removes overlapping UniProt IDs but does not filter by sequence similarity. This may allow high-similarity homologs to leak between train and test sets, leading to inflated performance estimates.

**Quality:**

3

**Strengths And Weaknesses:**

**Strengths**
1. The integration of ligand, holo pocket, and cavity representations enables improved robustness to pocket uncertainty.
2. Cross-attention-based cavity aggregation allows training on pocket-agnostic data, expanding applicability to real-world bioactivity datasets.
3. AANet achieves the best performance across the subset of DUD-E and LIT-PCBA.

**Weaknesses**
1. The paper does not provide information about the content and preprocessing steps of key datasets, such as ChEMBL35, used during training. This limits the transparency of the training process.
2. In Appendix Table S8, AANet underperforms DrugCLIP on the full LIT-PCBA dataset, despite showing strong results on curated subsets. This raises concerns about its generalization ability and suggests that the model may rely on cavity augmentation.

*Typo*
Line 285: “holoand”

---

> ### Author Rebuttal · Authors · 2025-07-31
>
> > W1: The paper does not provide information about the content and preprocessing steps of key datasets, such as ChEMBL35, used during training. This limits the transparency of the training process.
>
> We thank the reviewer for pointing this out. We will include the following details in the final version for clarity.
>
> The ChEMBL35 dataset was processed as follows:
>
> **Molecule filtering**:
>
> * Removed salts or kept only the largest fragment
> * Kept molecules with molecular weight between [100, 800]
> * Removed those containing atoms other than [H, C, N, O, F, Cl, Br, I, S, P, B, Se]
> * Filtered out molecules with unbranched long chains containing 6 atoms or more.
>
> **Activity filtering**: We retained activity records with:
>
> * Confidence score = 9
> * Assay type = functional or binding
> * Standard type ∈ ['Ki', 'IC50', 'Kd', 'EC50', 'ED50', 'AC50', 'XC50']
> * Values converted to molar units and filtered to lie within the –log10 range [5, 12].
>
> **Target mapping**: Protein targets were mapped to UniProt IDs and matched with AlphaFold structures. Targets without available predicted structures were discarded.
>
> **Contrastive supervision masking**: Each known protein–ligand activity pair was recorded using its UniProt ID and InChI-key. During training, we masked non-diagonal entries that correspond to active (positive) pairs to ensure that true positive pairs are never treated as negatives, even if they are not aligned in the current batch. This masking strategy ensures correct supervision and avoids misleading the model during contrastive learning.
>
> > W2: In Appendix Table S8, AANet underperforms DrugCLIP on the full LIT-PCBA dataset, despite showing strong results on curated subsets. This raises concerns about its generalization ability and suggests that the model may rely on cavity augmentation.
>
> We thank the reviewer for the observation. We would like to clarify the following:
>
> **Marginal difference and metric interpretation**:
> The performance gap is marginal (EF@1%: 5.51 vs 5.13), and our model **outperforms DrugCLIP in BEDROC** (0.0677 vs. 0.0623), which better reflects early enrichment — a critical criterion for practical screening.
>
> **Unconventional binding sites in LIT-PCBA**:
> LIT-PCBA contains atypical targets such as **mTORC1** and **PKM2**, where the ligand binds at **oligomeric interfaces** (hetero- or homo-dimers). Our method uses **monomeric AlphaFold structures** for second-stage training, which may reduce performance on these targets. Notably, our model **trained only on PDBbind still outperforms DrugCLIP on the full LIT-PCBA dataset** (EF@1%: 5.86 vs. 5.51; BEDROC: 0.0699 vs. 0.0623), suggesting the performance is not dependent on cavity augmentation.
>
> **Broader generalization**:
> Our method consistently outperforms DrugCLIP on a variety of datasets, including **DUD-E**, its curated subsets, **LIT-PCBA subsets**, and the newly added **DEKOIS benchmark**. This demonstrates strong generalization beyond any specific data augmentation strategy.
>
> **Table 1.** Performance on DEKOIS (77 targets) under holo, apo-pred-annotated (the cavity detected by FPocket that is closest to the holo pocket), and apo-pred-blind settings.
>
> | Method  | AUROC (holo) ↑ | AUROC (apo-annot) | AUROC (apo-blind) | BEDROC (holo) ↑ | BEDROC (apo-annot) | BEDROC (apo-blind) | EF\@0.5% (holo) ↑ | EF\@0.5% (apo-annot) | EF\@0.5% (apo-blind) |
> |-|-|-|-|-|-|-|-|-|-|
> | Vina | –  | – | –  | –| –| – | 2.58 | – | – |
> | GOLD | –  | – | –  | –| –| – | 10.32| – | – |
> | Glide | –  | – | –  | –| –| – | *11.85*| – | – |
> | DrugCLIP  | 0.7967  | 0.7075  | 0.4355 | 0.532| 0.322 | 0.0313 | 20.62| 12.12  | 0.92|
> | AANet | 0.8785  | 0.8624  | 0.7708 | 0.6446  | 0.5824| 0.3891 | 23.26| 20.91  | **13.33**  |
>
> > Q1: Is there any discussion of ligand-based virtual screening methods (LBVS)? For example, traditional LBVS methods (SHAFTS) and recent deep learning-based approaches (S-MolSearch) can bypass the need for protein structural information entirely.
>
> We thank the reviewer for highlighting the relevance of ligand-based virtual screening (LBVS), which can indeed bypass the need for protein structural information and is particularly useful when high-quality ligands are available or when protein structures are unreliable.
>
> We would like to clarify that **LBVS and SBVS address different use cases**. LBVS is often more straightforward when good query ligands are available (e.g., potent, exogenous ligands), but may be less effective in identifying novel chemotypes or in the absence of relevant prior ligands. In contrast, SBVS leverages the protein’s structural context, which can help discover structurally diverse actives but depends on structure quality.
>
> Regarding the methods mentioned:
>
> * SHAFTS is relatively outdated and unfortunately could not be executed on our current hardware.
>
> * S-MolSearch does not provide pre-trained checkpoints, training data, or inference code, making reproduction infeasible at this time.
>
> However, to address the reviewer’s concern, we **evaluated several widely-used LBVS methods**, covering both 2D and 3D paradigms, including **ECFP** (2D topology-based), **USR** (3D shape-based), and **Shape** from Schrödinger suite (3D shape overlap). The results are summarized in Table 2.
>
> **Table 2.** Performance of common LBVS methods on DUD-E benchmark
>
> | Method | EF@1% | BEDROC (α=80.5) |
> |-|-|-|
> | USR [1]  | 2.74  | 0.051 |
> | PhaseShape [2] | 8.29  | 0.138 |
> | ECFP4 [3] | 26.61 | 0.414 |
> | AANet (apo-blind) | 37.19 | 0.576 |
>
> [1] J Comput Chem. 2007, 28, 10, 1711-23.
> [2] J. Chem. Inf. Model. 2011, 51, 10, 2455–2466.
> [3] J. Chem. Inf. Model. 2010, 50, 5, 742–754.
>
> The notably strong performance of ECFP4 is likely due to the fact that many actives in virtual screening benchmarks such as DUD-E share similar topological scaffolds, whereas decoys are specifically designed to have similar physicochemical properties but divergent topologies. As a result, ECFP4 can effectively distinguish actives from decoys based on topological similarity alone, leading to higher enrichment factors. However, such advantages may not always generalize to real-world screening scenarios, where novel actives are often needed.
>
> > Q2: Is removing overlapping UniProt IDs sufficient to prevent information leakage between training and test sets? How do the authors ensure that models are not indirectly exposed to structurally or functionally similar targets?
> > L3: The data split removes overlapping UniProt IDs but does not filter by sequence similarity.
>
> We thank the reviewer for raising this important concern regarding potential overlap between training and test data.
>
> To further rule out information leakage, we performed an additional experiment using **MMseqs2 to remove any proteins in our training sets (PDBbind and ChEMBL) that contain chains sharing ≥90% sequence identity** with any DUD-E target. We then compared performance under both the original de-redundant setting and this stricter de-homologous setting. As shown in Table 3, **our method exhibits only a minor drop in performance**, maintaining strong results across all settings. This demonstrates the robust generalization of our approach, even in the absence of closely related training proteins.
>
> **Table 3.** Performance on DUD-E under de-redundant and de-homologous (90%) settings.
>
> | Method | BEDROC (holo) | BEDROC (apo-pred annot) | BEDROC (apo-exp annot) | BEDROC (apo-pred blind) | BEDROC (apo-exp blind) | EF@1% (holo) | EF@1% (apo-pred annot) | EF@1% (apo-exp annot) | EF@1% (apo-pred blind) | EF@1% (apo-exp blind) |
> |-|-|-|-|-|-|-|-|-|-|-|
> | DrugCLIP (de-redundant)   | 0.5157   | 0.3746    | 0.3493  | 0.1974    | 0.1926  | 33.70  | 22.70  | 21.36     | 12.05  | 11.75     |
> | DrugCLIP (de-homologous)  | 0.3949   | 0.2323    | 0.2134  | 0.1097    | 0.1298  | 24.90  | 13.56  | 12.59     | 6.51   | 7.36 |
> | AANet (de-redundant) | **0.6365**   | **0.6003**    | **0.5866**  | **0.6232**    | **0.5764**  | **40.85**  | **38.46**  | **38.03**     | **40.85**  | **37.19**     |
> | AANet (de-homologous)     | *0.5985*   | *0.5247*    | *0.5475*  | *0.5955* | *0.5411*  | *38.55*  | *33.59*  | *34.76*     | *38.34*  | *34.50*     |
>
> > L1: The model relies on external cavity detection (e.g., Fpocket). If the cavity predictions are poor—especially in complex or flexible proteins—the model’s performance may degrade.
>
> We thank the reviewer for raising this important point.
>
> 1. **Robustness across cavity detection tools:**
> Our method does **not rely on any specific cavity detection tool**. As shown in Table 4, AANet demonstrates **consistent performance across different pocket detection methods**.
>
> 2. **Limitation in flexible or disordered proteins**:
> We acknowledge that for highly flexible or intrinsically disordered proteins, where well-formed pockets are absent and ligand-induced fit is essential, **existing pocket-based SBVS methods, including ours, face inherent challenges**. Our work primarily addresses uncertainty in binding site localization, not full flexible binding modeling.
>
> We agree this is a critical direction, and we consider **end-to-end pocket prediction and flexible modeling** promising avenues for future research.
>
> > L2: No confidence or uncertainty estimation in predictions.
>
> We appreciate the reviewer’s insightful comment and agree that quantifying prediction confidence would enhance the method’s practical value. While AANet does not currently produce explicit uncertainty estimates, its design of modeling interactions with both binding and non-binding cavities naturally provides multiple levels of structural signal that could support future confidence modeling. Leveraging this structure-aware framework for uncertainty estimation is one of our priority directions for future work.

---

> > ### Comment · Reviewer_TXmb · 2025-08-05
> >
> > Thank you for the comprehensive and thoughtful rebuttal. The authors have clarified the key concerns I raised.
> > Based on the clarifications and new results, I am raising my score.

---

### Official Review · Reviewer_u1UJ · 2025-07-03

**Clarity:** 3
**Significance:** 2
**Originality:** 3
**Rating:** 4
**Confidence:** 2

**Summary:**

This paper introduces AANet, a novel virtual screening framework designed for structure-based drug discovery under structural uncertainty, particularly when working with apo or predicted protein structures lacking experimentally resolved binding pockets. AANet combines two key components: (1) a tri-modal contrastive alignment module that learns to align ligands, holo pockets, and geometry-detected cavities, and (2) a cross-attention-based aggregation module that dynamically scores and selects binding-relevant pockets from multiple candidates. The model is shown to significantly outperform both docking-based (Glide) and modern deep learning baselines (DrugCLIP, TankBind) in challenging blind apo settings, achieving near-holo performance. The authors also curate a benchmark derived from DUD-E and LIT-PCBA to evaluate under various pocket and structure scenarios.

**Questions:**

Can the authors clarify whether any target-level or structure-level overlap exists between training and test sets?

**Ethical Concerns:**

["NO or VERY MINOR ethics concerns only"]

**Final Justification:**

The authors have convincingly addressed the main concerns raised in the initial review.

Specifically, the addition of DEKOIS-based evaluation (Table 1) strengthens the evidence for generalizability beyond the original benchmarks. The explanation of why FlowDock is less suited for large-scale virtual screening, along with the new empirical comparison (Table 2), provides a clear justification for its exclusion as a primary baseline. The clarification on cavity detection and the demonstration of robustness across tools further support the model's design choices.

Given these clarifications and additional results, I am increasing my score.

**Quality:**

3

**Strengths And Weaknesses:**

[Strengths]

* The combination of tri-modal contrastive alignment (between ligand, holo pocket, and detected cavity) and cross-attention aggregation is new and hasn't been used in previous methods.

* The method shows large improvements compared to both traditional docking tools and recent deep learning approaches—especially in hard cases like blind apo structures.

* The paper includes careful ablation studies and tests with different cavity detection tools, showing that the model is robust and generalizes well.

[Weaknesses]

* Even though the results are strong on DUD-E and LIT-PCBA, the method is not tested on more diverse or unbiased datasets, such as DEKOIS, or in real-world drug discovery tasks.

* The model relies on external tools like Fpocket to find candidate pockets, which prevents end-to-end learning and may cause biases depending on the tool used.

* The paper does not compare to some of the latest state-of-the-art methods, such as FlowDock or ProFSA, which also aim to handle apo structures under uncertainty. Notably, FlowDock includes an end-to-end pocket detection capability, making it more convenient to use. This highlights the need for a direct comparison between the two methods.

---

> ### Author Rebuttal · Authors · 2025-07-31
>
> > W1: Even though the results are strong on DUD-E and LIT-PCBA, the method is not tested on more diverse or unbiased datasets, such as DEKOIS, or in real-world drug discovery tasks.
>
> We thank the reviewer for pointing this out. Due to time constraints, we were unable to identify and verify the experimental apo structures corresponding to DEKOIS targets. However, we mapped the DEKOIS targets to UniProt ID and successfully downloaded 77 target structures from the AlphaFold Database, excluding 4 virus-related targets that are not available (HIV1PR & HIV1RT: P03366, SARS-HCoV: P0C6X7, NA: P27907).
>
> We evaluated our method and baselines under three settings:
>
> * **Holo**: using the known holo pocket.
>
> * **Apo-pred (annotated)**: pocket detected near the ligand (used only for annotation).
>
> * **Apo-pred (blind)**: fully blind screening setting.
>
> As shown in Table 1, our method consistently **outperforms DrugCLIP across all settings**, and **surpasses traditional docking baselines under holo pocket settings** (*italic*) when tested under **the most challenging apo-pred blind setup** (**bolded**).
>
> **Table 1.** Performance on DEKOIS (77 targets) under holo, apo-pred-annotated (the cavity detected by FPocket that is closest to the holo pocket), and apo-pred-blind settings.
>
> | Method    | AUROC (holo) ↑ | AUROC (apo-annot) | AUROC (apo-blind) | BEDROC (holo) ↑ | BEDROC (apo-annot) | BEDROC (apo-blind) | EF\@0.5% (holo) ↑ | EF\@0.5% (apo-annot) | EF\@0.5% (apo-blind) |
> |-|-|-|-|-|-|-|-|-|-|
> | Vina | –  | –   | –    | –| –| – | 2.58 | – | –   |
> | GOLD | –  | –   | –    | –| –| – | 10.32| – | –   |
> | Glide     | –  | –   | –    | –| –| – | *11.85*| – | –   |
> | DrugCLIP  | 0.7967  | 0.7075  | 0.4355   | 0.532| 0.322 | 0.0313 | 20.62| 12.12  | 0.92|
> | AANet     | 0.8785  | 0.8624  | 0.7708   | 0.6446    | 0.5824| 0.3891 | 23.26| 20.91  | **13.33**    |
>
>
> > W2: The model relies on external tools like Fpocket to find candidate pockets, which prevents end-to-end learning and may cause biases depending on the tool used.
>
> We thank the reviewer for this insightful comment. While our current framework employs Fpocket for cavity detection during training, we would like to clarify and justify this design with the following points:
>
> 1. **Robustness across different cavity detectors**: Although our model uses Fpocket to extract training cavities, **the downstream performance remains stable across different cavity detection tools** (cf. Table 4). These conventional detectors are grounded in **physicochemical definitions of binding pockets**, such as surface concavity, void volume, and geometric constraints. The consistency across tools suggests that AANet is **not overfitting to a specific detection algorithm**, but instead learns **underlying structural priors** intrinsic to proteins, such as spatial enclosures or clefts.
>
> 2. **Alignment with practical structure-based pipelines**: Our method follows the typical structure-based virtual screening pipeline, where **pocket detection and binding assessment are distinct stages**. This modularity aligns with standard workflows in drug discovery and allows our method to **plug into existing infrastructure** while remaining adaptable to alternative cavity detectors.
>
> 3. **Physically meaningful supervision**: By grounding learning on cavities defined via physical heuristics (e.g., void detection), we introduce inductive bias from medicinal chemistry intuition into the model. This helps prevent overfitting to spurious binding patterns and encourages the model to focus on **biophysically** plausible pockets. **Concerns about physical plausibility have recently gained attention** in the structure prediction and docking communities, as highlighted by works such as PoseBusters in deep learning-based docking. This emphasis on biophysical realism is equally relevant to virtual screening and motivates the design of our method.
>
> Although we currently do not explore end-to-end cavity prediction due to time and resource constraints, we agree that **learning pocket detection in an end-to-end fashion is a promising direction**, especially on novel targets including flexible proteins. As datasets grow and modeling techniques evolve, we believe this integration could offer even more robust and generalizable solutions.
>
> > W3: Lack of comparison with recent methods like FlowDock and ProFSA.
>
> While we appreciate the reviewer’s suggestion regarding FlowDock and ProFSA, which enrich the diversity of baseline methods, we would like to clarify that **flexible docking frameworks like FlowDock are not primarily designed for large-scale virtual screening tasks**. Instead, they are better suited for **post-screening re-ranking or refinement stages**, where a small number of candidate actives are already identified.
>
> Flexible docking excels at capturing subtle conformational changes and fine-grained interactions between a specific ligand and a pocket, making it valuable for discriminating between a few similar binders. However, in large-scale virtual screening where the key objective is to enrich true actives from a vast set of mostly inactive molecules, such methods are **less effective when screening against millions of decoys**, as required in virtual screening settings. We will **include a discussion of flexible docking methods like FlowDock in the Related Work section** of the paper to provide a more complete picture in the context of apo setting.
>
> To address the reviewer’s concern, we included results from DrugCLIP (ProFSA) using the checkpoint publicly released in the supplementary data of bioRxiv 2024.09.02.610777, which was fine-tuned from ProFSA with the DrugCLIP task. Interestingly, DrugCLIP (ProFSA) underperforms the original DrugCLIP baseline in our virtual screening benchmark. We hypothesize that this may be due to its pseudo-ligand-based pretraining, which may have enhanced the model’s ability to recognize general protein–ligand interaction patterns but also made it more prone to misinterpreting peptide-binding interfaces as small-molecule binding pockets.
>
> Although FlowDock’s computational cost prevented us from performing large-scale comparisons on the full dataset, we conducted a direct empirical evaluation on a representative subset. Specifically, we randomly sampled **10% of actives and 5% of decoys across 38 apo targets from DUD-E**, which may approximately **halve the number of Enrichment Factors (EF)** reported. For efficiency, we **generated only one ligand conformer per molecule** (instead of the default five), which **may degrade FlowDock’s performance**. This reduced the total number of complexes to < 30k and allowed us to complete all FlowDock runs within 3 GPU days. These results are shown in Table 2.
>
> In contrast, our method directly targets binder enrichment under structural uncertainty and is optimized for efficiency, robustness, and early enrichment metrics. Nevertheless, AANet significantly outperforms FlowDock across all key metrics on the subset, as shown below. We also acknowledge FlowDock’s strengths in generating high-quality complex structures, which may provide additional interpretability or utility in downstream stages of the drug discovery pipeline.
>
> **Table 2.** Performance comparison on apo-blind virtual screening subset.
>
> | Method | AUROC ↑ | BEDROC (α=80.5) ↑ | EF@1% ↑ | EF@5% ↑ |
> |-|-|-|-|-|
> | DrugCLIP    | 0.6342 | 0.2567 | 9.04   | 4.58   |
> | DrugCLIP (ProFSA)| 0.6317 | 0.1799 | 5.71   | 3.83   |
> | FlowDock    | 0.5863 | 0.2358 | 8.15   | 5.24   |
> | AANet (Ours) | **0.8651** | **0.6729** | **24.38** | **12.12** |
>
> > Q1: Can the authors clarify whether any target-level or structure-level overlap exists between training and test sets?
>
> We thank the reviewer for raising this important point.
> On **PDBbind**, we removed duplicate complexes based on structure IDs, following prior work to ensure fair comparison. However, this may not fully eliminate target-level redundancy. For some of baselines, duplicates were not removed (cf. Supplementary Section B.7). On **ChEMBL**, we filtered data using UniProt IDs, which more reliably removes overlapping targets.
>
> To further address potential leakage, we conducted an additional experiment by using MMseqs2 to **identify and remove any training proteins (from PDBbind and ChEMBL) containing chains with ≥90% sequence identity** to any DUD-E structure / target sequences. We then compared our method and DrugCLIP under both the original and this more stringent de-redundant / de-homologous setting. As shown in Table 3, our method maintains robust performance even after removing homologous sequences, demonstrating better generalization.
>
> **Table 3.** Performance on DUD-E under de-redundant and de-homologous (90%) settings.
>
> | Method | BEDROC (holo) | BEDROC (apo-pred annot) | BEDROC (apo-exp annot) | BEDROC (apo-pred blind) | BEDROC (apo-exp blind) | EF@1% (holo) | EF@1% (apo-pred annot) | EF@1% (apo-exp annot) | EF@1% (apo-pred blind) | EF@1% (apo-exp blind) |
> |-|-|-|-|-|-|-|-|-|-|-|
> | DrugCLIP (de-redundant)   | 0.5157   | 0.3746    | 0.3493  | 0.1974    | 0.1926  | 33.70  | 22.70  | 21.36     | 12.05  | 11.75     |
> | DrugCLIP (de-homologous)  | 0.3949   | 0.2323    | 0.2134  | 0.1097    | 0.1298  | 24.90  | 13.56  | 12.59     | 6.51   | 7.36 |
> | AANet (de-redundant) | **0.6365**   | **0.6003**    | **0.5866**  | **0.6232**    | **0.5764**  | **40.85**  | **38.46**  | **38.03**     | **40.85**  | **37.19**     |
> | AANet (de-homologous)     | *0.5985*   | *0.5247*    | *0.5475*  | *0.5955* | *0.5411*  | *38.55*  | *33.59*  | *34.76*     | *38.34*  | *34.50*     |

---

> > ### Comment · Reviewer_u1UJ · 2025-08-01
> > **re**
> >
> > Thank you for the detailed and thoughtful rebuttal. The authors have convincingly addressed the main concerns raised in the initial review.
> >
> > Specifically, the addition of DEKOIS-based evaluation (Table 1) strengthens the evidence for generalizability beyond the original benchmarks. The explanation of why FlowDock is less suited for large-scale virtual screening, along with the new empirical comparison (Table 2), provides a clear justification for its exclusion as a primary baseline. The clarification on cavity detection and the demonstration of robustness across tools further support the model's design choices.
> >
> > Given these clarifications and additional results, I am increasing my score.

---

### Official Review · Reviewer_ETw8 · 2025-07-05

**Clarity:** 3
**Significance:** 4
**Originality:** 4
**Rating:** 5
**Confidence:** 3

**Summary:**

This paper tackles the challenge of virtual screening on apo or predicted protein structures, where binding pocket locations are unknown. The authors propose AANet, a two-phase framework to address this structural uncertainty. First, a tri-modal contrastive learning scheme aligns representations of ligands, holo pockets, and detected cavities to learn robust features. Second, a cross-attention adapter aggregates information from multiple candidate cavities to infer the most likely binding site. Through extensive experiments, the authors demonstrate that AANet significantly outperforms state-of-the-art methods in this challenging blind screening setting, achieving performance comparable to that on ideal, ligand-bound structures.

**Questions:**

1. The tri-modal contrastive learning is central to the proposed method. In this framework, the model is required to align with both the ideal 'holo pocket' and the 'detected cavity'. My question is: given that the ultimate goal is to predict on apo structures, what is the key role of aligning with the 'holo pocket'? Is this step primarily intended to provide a stable learning anchor to prevent the model from deviating on noisy data, or are there other, deeper considerations? This would help readers understand the intrinsic logic of the framework's design.

**Ethical Concerns:**

["NO or VERY MINOR ethics concerns only"]

**Final Justification:**

The authors have addressed my concerns during the rebuttal. In addition, based on the comments of other reviewers, I'd like to keep the positive score.

**Paper Formatting Concerns:**

There are no major formatting issues.

**Quality:**

4

**Strengths And Weaknesses:**

Strengths:
1. Addresses a timely and critical problem: The paper focuses on how to effectively utilize apo structures from tools like AlphaFold2 for virtual screening, a key bottleneck in current structure-based drug discovery. This work provides a viable solution for leveraging vast amounts of AI-predicted structural data, making it highly significant and impactful in a real-world context.
2.  Elegant Methodological Design: The AANet framework, particularly its tri-modal contrastive alignment and ligand-conditioned cross-attention aggregation mechanisms, presents a highly original and well-tailored design to address the problem of pocket uncertainty.
3. Thorough and Rigorous Experiments: The experimental design is highly rigorous. Notably, the authors constructed a benchmark with matched holo, experimental apo, and predicted apo structures, and designed multiple settings ('Oracle,' 'Annotated,' and 'Blind') to clearly dissect the sources of the method's advantages.

Weaknesses:
1. Lack of discussion on computational efficiency: The paper's core strength lies in its excellent performance, but it does not provide a quantitative analysis of its computational cost. A comparison of runtime between AANet and baselines like DrugCLIP would offer a more complete picture of the accuracy-efficiency trade-off, which is important for assessing its practicality in large-scale screening.
2. The justification for some methodological parameters could be clarified: The authors mention increasing the pocket extraction radius from 6 Å to 10 Å to improve performance. While the ablation study demonstrates the effectiveness of this change, the choice of 10 Å appears somewhat empirical. A more detailed explanation of the selection criteria for such a key parameter would help readers better understand the method.

---

> ### Author Rebuttal · Authors · 2025-07-31
>
> > W1: Lack of discussion on computational efficiency
>
> To address the reviewer's concern, we provide a direct comparison of overall computational cost among representative methods on the LIT-PCBA dataset, using a single NVIDIA A100 80GB GPU and a 128 vCPU server.
>
> | Method     | Sharing Cost*                          | PCBA-One** (2.8M) | PCBA-Full (10M) | PCBA-Apo-Blind (100M) | Overall Cost         |
> |------------|----------------------------------------|------------------|------------------|-------------------------|-----------------------|
> | Glide      | LigPrep: 1 hr (930 mols/s)                     | 3 days           | 11 days          | 3.6 months              | 3 days – 3.6 months   |
> | RTMScore  | – | Preprocessing: 3 hrs (247/s) + Inference 8.2 hrs (94/s) | 11.2 hrs +  29.5 hrs | 17 days | 11.2 hrs - 17 days |
> | EquiScore | – | Preprocessing: 2.7 hrs (291/s) + Inference 8.1 hrs (96/s) | 11 hrs + 29 hrs | 16.7 days | 10.8 hrs - 16.7 days |
> | TankBind   | – | 7 hrs              | 25 hrs             | 10.4 days               | 7 hrs – 10.4 days       |
> | DrugCLIP   | Mol: 270 s (10,370/s), Pocket: 1–2 s (86/s) | –                | 15 s              | 15 s                     | ~300 s               |
> | AANet      | Mol: 270 s (10,370/s), Pocket: 1–3 s (36/s) | –                | 15 s              | 15 s                     | ~300 s               |
>
> \*: Cost that may be shared across subsequent virtual screening (VS) projects;
> \*\*: One highest-resolution PDB structure was selected for each subset (cf. Supplementary Section B.6).
>
> As our dual-tower backbone does not involve fusion prior to the adapter and supports embedding reuse, our method (AANet) introduces minimal overhead beyond DrugCLIP while providing significantly improved performance, particularly in unbound settings. The majority of runtime during the scoring phase arises from data loading and CPU–GPU communication, rather than model inference itself. **On typical commercial-scale libraries, AANet maintains ultra-fast screening throughput** comparable to DrugCLIP.
>
> > W2: Clarification on the 10 Å pocket extraction radius
>
> We appreciate the reviewer’s concern regarding the selection of the 10 Å threshold for pocket extraction. This choice was based on quantitative analysis on the PDBbind dataset, supported by the following three key findings:
>
> 1. **Coverage of original holo pockets**: The oracle 10 Å cavities, which are defined as those with the highest overlap (IoU) with holo pockets, **fully cover 75.38% of the original 6 Å holo pockets, and 90.94% achieve ≥ 85% coverage**. This indicates that 10 Å cavities provide a reliable approximation of the ground-truth pocket regions.
>
> 2. **Tradeoff between feature integrity and computational cost**: The average atom count in 10 Å oracle cavities is **612.68 ± 240.75, over 3× larger** than the 6 Å holo pockets (199.32 ± 55.04). Since computational cost scales non-linearly with the number of input atoms, we **uniformly downsample pockets to ≤ 256 atoms** during training to ensure **fair comparisons across different pocket definitions**. Larger regions would require more aggressive downsampling, which may **compromise spatial features or increase computational overhead**. Thus, 10 Å represents a practical upper bound that balances representational fidelity with efficiency.
>
> 3. **Avoiding bias from over-extension**: Extending the radius beyond 10 Å **risks incorporating off-target cavities**, especially on cluttered protein surfaces. As our 10 Å is defined by minimal atom-to-atom distance (not center-based), it already captures ample structural context. Further expansion would dilute the binding signal and potentially introduce irrelevant patterns.
>
> **Thus, the 10 Å threshold strikes a practical balance**: it ensures high coverage of the ground-truth binding pocket, preserves structural integrity, and avoids unnecessary computational overhead or learning from irrelevant patterns.
>
> > Q1: What is the role of aligning with the 'holo pocket' in tri-modal contrastive learning?
>
> We thank the reviewer for this insightful question. The alignment with holo pockets plays a critical and **bidirectional** role in our tri-modal contrastive framework, motivated by the following:
>
> 1. **Holo → Cavity - Guidance for learning from noisy input**:
> Holo pockets, shaped by actual ligand binding, serve as **reliable supervisory anchors** (as stated by the reviewer). Aligning detected cavities with holo pockets helps the model extract meaningful features even from noisy or imprecise cavity detection.
>
> 2. **Cavity → Holo - Addressing structural uncertainty and generalization**:
> Holo pockets are **biased toward a specific ligand series**(cf. Suppl. A.1-3), while other actives may bind elsewhere. By aligning cavities (which are ligand-agnostic) to holo pockets, the model learns features that are **spatially intrinsic to the protein**. This improves the model’s robustness to structural uncertainty and enhances generalization. This alignment also **boosts performance on holo structures**, since the learned features are less tied to ligand-specific conformations.
>
> 3. **Cavity ↔ Holo - Dual modeling of general and specific signals**:
> The model jointly learns **ligand-independent (intrinsic to proteins) and ligand-specific** pocket representations, which improves robustness. This enables **stronger performance across both holo and apo structures**, not just mitigating the performance gap but outperforming DrugCLIP in both cases.
>
> **In summary**, holo-cavity alignment is essential for modeling **both precise and generalizable binding patterns**, supporting our goal of robust virtual screening under structural uncertainty.

---

> > ### Comment · Reviewer_ETw8 · 2025-08-09
> >
> > Thank you for your detailed responses which have addressed my concerns. I have no more questions and will keep the positive score.

---

### Comment · Area_Chair_auUh · 2025-08-05

Dear reviewers,

Thanks for your contribution to the reviews, and also thanks for those who have make responses yet.

The authors have put rebuttal to the reviews, please look at the rebuttal and make responses accordingly. Note that engaging the discussion is important at this phase.

Please make sure that you give necessary comments to the rebuttal and then make acknowlegement, therefore the authors can have a better understanding whether their rebuttal help to your concerns.

Best, AC

---

### Decision · Program_Chairs · 2025-09-17

**Decision:**

Accept (poster)

**Comment:**

This paper focuses on the apo-structure protein binding with ligand for virtual screening task. The main contribution of the paper is that it proposes a tri-modal contrastive learning method and the a cross-attention based adapter training. Though advanced experiments, the method shows improvement on the early enrichment factory by a large margin.

Reviewers have give positive feedback to the work, on the novelty of the method and the reasonable setting and clear motivation. The results are also well acknowledged by the reviewers. Though some concerns have raised, most of them have been addressed and the reviewers are mostly satisfied of the rebuttal.
The reviewers are encouraged to add more discussion as responded in the rebuttal.